# Designing an Environmental Wind Tunnel Cooling System for High-Speed Trains with Air Compression Cooling and a Sensitivity Analysis of Design Parameters

**DOI:** 10.3390/e25091312

**Published:** 2023-09-08

**Authors:** Junjun Zhuang, Meng Liu, Hao Wu, Jun Wang

**Affiliations:** School of Aeronautic Science and Engineering, Beihang University, Beijing 100191, China; liumeng@buaa.edu.cn (M.L.); haowu@buaa.edu.cn (H.W.); wangjun@buaa.edu.cn (J.W.)

**Keywords:** cooling system, one-dimensional analysis method, principle of similarity, sensitivity analysis

## Abstract

Environmental wind tunnels for high-speed trains play a significant role in their development. The cooling system of the wind tunnel poses a challenge as it requires lower temperatures and a higher cooling capacity during operation. The conventional approach to wind tunnel refrigeration uses evaporative cooling, which is less efficient at low temperatures and comes with environmental and safety risks. In this study, we propose an innovative air compression refrigeration method based on the Brayton cycle. This method converts high-pressure air into low-temperature air at atmospheric pressure for wind tunnel refrigeration. The new cooling system has reduced energy usage by 3.72 MW, leading to a 13.15% improvement. The return cooler of the system is modeled using the effective number of heat transfer units and the mean temperature difference design method. Additionally, the turbine within the system is analyzed using one-dimensional flow characteristic analysis and the principle of similarity. This method has been validated by comparing it to other published papers. Subsequently, we perform a thorough sensitivity analysis on the key design parameters of the system. We observe that with a sufficient heat transfer area of the recooler, the cooling efficiency of the system exhibits a gradual decline from 64% to 60% as the mass flow rate of the system rises. For a fixed turbine, the cooling efficiency of the system rises from 20% to 62% and subsequently declines to 37%, with an increase in the mass flow rate. As a result, we conclude that the design parameters of the turbine have a more significant influence on the cooling efficiency of the system than the recooler. Our study will establish a foundation for selecting parameters to optimize the refrigeration system in the future.

## 1. Introduction

Rail transport has become an indispensable part of transportation systems worldwide due to its speed and stability. In recent years, high-speed railway technology has matured and developed, elevating the importance of high-speed rail, particularly in countries like China, Japan, and Germany, where it was introduced earlier. However, the progress of technologies related to high-speed trains has been impeded by the lack of experimental facilities, specifically those that simulate the extreme temperature conditions experienced by these trains.

Currently, only two laboratories, namely the Arsenal International Vehicle Test Station in Austria and the Kawasaki Heavy Industries Vehicle Environment Room in Japan, can conduct low-temperature experiments on trains. However, these facilities are unable to fully replicate the external conditions encountered by high-speed trains during operation, often restricting low-temperature experiments to natural outdoor conditions. This limitation significantly hampers the advancement of high-speed train technologies. Consequently, the need arises to establish a full-size high-speed railway train environmental wind tunnel to enhance the efficiency and capabilities of high-speed train experiments.

An environmental wind tunnel for high-speed trains must accurately simulate various train operating conditions under natural circumstances. It should have the ability to replicate different temperatures and wind speeds. Given that the operation of the wind tunnel itself generates a substantial amount of heat, controlling its dissipation becomes crucial when simulating a high-temperature environment for experimentation. On the other hand, creating a low-temperature environment is challenging, especially when simulating the conditions of high-speed trains running at extremely low temperatures. In such cases, not only must the temperature within the experimental section of the wind tunnel be reduced to below −70 °C, but efficient heat management during operation is also essential. Therefore, selecting an appropriate wind tunnel cooling system becomes a vital aspect of designing and constructing an environmental wind tunnel for high-speed trains.

Most existing environmental wind tunnels for high-speed trains utilize evaporative cooling. However, evaporative cooling has certain drawbacks, especially at low temperatures, where its efficiency is reduced. In some cases, auxiliary cooling methods, like liquid nitrogen are necessary to achieve lower ambient temperatures. Additionally, the evaporative refrigeration cycle requires the use of refrigerants, which not only pose safety and environmental pollution risks, but also increase equipment maintenance costs. On the other hand, the air compression–refrigeration cycle offers high reliability and ease of maintenance. In this experiment, normal dry air is used as the refrigerant, eliminating any possibility of environmental contamination. The application of air compression refrigeration has been increasingly used in ultra-low-temperature refrigeration and cooling, as well as in air conditioning, aircraft, automotive, and various other fields [1].

The air refrigeration system used in this study was designed by Normalair Garrett Limited, a British company. They have developed a high-speed railway train air conditioning system that employs a semi-open, two-stage compressed, and boosted air refrigeration system [2]. A.J. White applied the inverse Brayton air cooling cycle to a room air conditioning system and compared its performance with and without heat rejection. The results show that increasing the heat exchanger efficiency significantly improved the system’s overall performance [3,4]. Air compression refrigeration systems are also used in the civil aircraft process, and a near-logarithm heat transfer temperature difference function is presented [5]. Furthermore, the performance of the inverse Brayton heat pump system was simulated under different operating conditions, and the differences in performance between storage and instantaneous heat modes of various heat pumps were analyzed. Zhang Han et al. [6,7,8] achieved electric–thermal and cooling conversion through the reverse Brayton cycle, thus enabling energy storage. Zhang [9] analyzed a 100,000 m^3^/h air compression system and explored methods to enhance its power [9]. Recent research on air compression refrigeration systems has resulted in improved efficiency, indicating that air compression refrigeration systems might be a viable replacement for evaporative refrigeration systems in environmental wind tunnels.

In recent years, S. K. Park et al. [10] conducted simulation studies on air compression refrigeration to numerically simulate the performance of open-air refrigeration systems under non-designed conditions. Similarly, A. K. Dhillon et al. [11] optimized the inverse Brayton cycle system using efficiency as the objective function, providing guiding principles for selecting optimal operating parameters under different temperature conditions. The air refrigeration system features a simple configuration with key components, including a turbine expander, compressor, and heat exchanger. The expander plays a central role in air compression refrigeration, significantly influencing the system’s performance [12]. Liu [13] also investigated the effects of several operational parameters on the performance of the Organic Rankine Cycle (ORC) system, such as the mass flow rate of the working fluid, the mass flow rate of air in the condenser, turbine inlet pressure, and inlet temperature, aiming to achieve maximum net power output. Historically, air turbine efficiencies were often modeled as constant values; however, due to the diverse air flow rates required by high-speed train environmental wind tunnel air cooling systems, a constant efficiency model fails to accurately simulate turbine efficiencies under varying operating conditions. Consequently, Song et al. [14] proposed a one-dimensional model to predict the efficiency of a radial inflow turbine in an ORC. They subsequently conducted a parametric analysis of the system along with the one-dimensional turbine efficiency. Xia et al. [15] further delved into this method, employing a three-dimensional model to analyze the associated outcomes. Niu Lu et al. [16] developed a mathematical approach to predict the performance of a turbine expander under off-design conditions, analyzing the impact of pressure ratio, inlet temperature, and rotational speed on performance. The predicted results align well with experimental data. Ke Changlei et al. [17] conducted numerous numerical simulations on a high-speed mixed-flow centripetal turbine under various design and cooling conditions (e.g., inlet pressure and brake power). By achieving flow capacity matching under these conditions, the method demonstrated superior performance prediction capabilities for low-temperature turbines. The air compressor is another pivotal component in air compression refrigeration systems, with energy-flow coupling characteristics akin to those of the expander. However, previous studies have focused on the performance of a single design size turbine, and rarely address how turbines of various sizes perform. El et al. [18] performed Exergy Analysis of a Turbo Expander using Matlab to investigate the performance of turbines of different sizes. This approach, however, necessitates a significant amount of experimental data and is not conducive to system design.

Recoolers and water coolers are typically employed in air compression refrigeration systems to cool the high-temperature gas. Due to recirculation without any mass phase changes and the air’s low constant-pressure specific heat capacity, compact heat exchangers are preferred. Plate heat exchangers find broad application across multiple domains and systems owing to their compact structure; high heat transfer efficiency; flexible combination of heat transfer surfaces; and ease of installation, disassembly, and cleaning [19,20,21]. They are the preferred choice for water coolers and return coolers in air compression refrigeration systems. Marinheiro [22] introduced plate heat exchangers into the Rankine cycle, calculating their heat transfer efficiency and other parameters within the refrigeration cycle. Pouyan [23] investigated the relationship between the average temperature of the heat transfer fluid and the inlet flow rate for different plate heat exchanger structures. Additionally, Arcíadealva [24] applied plate heat exchangers to a solar absorption cooling system, verifying the operating parameters of the heat exchangers under diverse conditions. The performance simulation of plate heat exchangers has been extensively studied. Researchers such as Xu [25,26] have employed numerical simulations to analyze the heat transfer and resistance characteristics of plate heat exchangers. RIOS-IRIBE [27] further explored the effects of the number of plates and plate spacing on heat transfer and pressure drops using a CFD model. Experimental validation was performed based on the simulation results, confirming the reliability of the model. This approach boasts greater precision, but it presents challenges in integrating with other models when simulating the refrigeration system. As a remedy, we will establish a numerical model to enhance computational efficiency while also incorporating various design parameters into the plate heat exchanger’s performance within the system. In this paper, we aim to establish a model that incorporates the design parameters of a plate heat exchanger for analyzing the performance of the exchanger as a recooler in the system.

This paper aims to introduce a novel refrigeration system, namely the air compression refrigeration system, which differs from the conventional evaporative refrigeration system used in wind tunnels. The system pressurizes dry air into high-temperature, high-pressure air and utilizes the Brayton cycle to convert it into atmospheric low-pressure gas suitable for cooling the system. Subsequently, the low-temperature gas is directly discharged into the wind tunnel, where it mixes with the original gas to achieve cooling. In previous simulations of air compression refrigeration systems, studies on turbines failed to include the analysis of turbines with different sizes, resulting in difficulties in optimizing the system selection and elevating the probability of suboptimal optimization. To tackle this dilemma, we aim to utilize one-dimensional flow modeling and similarity principles in our study. We will establish a dynamic model of the constituents dependent on earlier research. Using this, a model of the air compression refrigeration system will be created. Through sensitivity analysis of the components, we identify the factors with the greatest influence on air compression refrigeration system efficiency, thereby paving the way for further research.

## 2. Thermodynamic Model for Air Compression Refrigeration Systems

Prior to diving into the design intricacies of the high-speed train environmental wind tunnel, it is crucial to contemplate the cooling requirements of the wind tunnel. The essential aspects can be understood from Figure 1, which illustrates the cooling demands of the refrigeration system, mainly focusing on temperature specifications and cooling capacity prerequisites. It is important to note that the temperature requirements are primarily influenced by the thermal criteria set by the experimental segment dedicated to high-speed trains within the wind tunnel. Furthermore, since the wind tunnel’s cooling system is responsible for providing optimal cooling for all the equipment housed inside, the cooling needs of the air compression system are closely intertwined with the operational status of the mentioned equipment. During the peak velocity operations of the wind tunnel, it is important to recognize the power section as a highly energy-intensive subsystem within the facility. The kinetic energy harnessed by this section is completely dissipated within the wind tunnel, making it an indispensable aspect of the overall cooling requirements. In addition, it is important not to neglect the heat transfer requirements of other systems, including electrical systems, lighting systems, and life support systems. Therefore, the carefully designed air compression cooling system presented in this article aims to extract ambient air from the wind tunnel itself. This approach effectively mitigates the energy consumption associated with the power section’s substantial energy demands.

In the aerodynamic wind tunnel for high-speed trains, an air compression refrigeration system is employed, as shown in Figure 2. The system, based on the Brayton cycle, compresses and expands air to generate low-temperature refrigerated air for cooling the wind tunnel. The main components include a inlet compressor, turbo compressor, turbo expander, water cooler, return cooler, and cooling tower. The specific workflow is as follows: External air is compressed and undergoes drying and purification through the inlet compressor, resulting in high-temperature and high-pressure gas. It is then further pressurized by the turbo compressor and cooled through the cooling system. Finally, it enters the turbo expander, transforming into low-temperature gas at atmospheric pressure, which serves as the cold source for the wind tunnel. The work output of the expander can be recovered to improve the system’s energy efficiency. Additionally, excess hot gases are expelled through an air pump to maintain stable pressure within the wind tunnel. These expelled gases not only reduce wind resistance in the wind tunnel, but also simulate the ground suction process in the high-speed train environment.

The air compression refrigeration system typically employs centrifugal compressors or screw compressors, both driven by electric motors or used in conjunction with turbines to enhance air pressure. The compression ratio, a vital parameter of the compressor, can be expressed as follows:$$\varepsilon_{c}=\frac{p_{out}}{p_{in}}$$
Assuming that the compressor’s compression process approximates an isentropic adiabatic compression, the outlet temperature of the compressor is given by
$$T_{out}=T_{in}\left(\frac{\varepsilon_{c}{}^{\frac{k-1}{k}}-1}{\eta_{c}}+1\right)$$
where $\eta_{c}$ represents the adiabatic efficiency of the compressor, denoting the ratio of isentropic compression work to actual power consumption. Assuming there is minimal, and thus negligible, gas leakage within the compressor and that the mass flow rate of the gas remains constant between the inlet and outlet, the required power for the compressor is determined as follows:$$P_{c.out}=\dot{m}c_{p}\left(T_{out}-T_{in}\right)$$
The following formula is used to calculate the unit cooling capacity of gas refrigeration equipment:$$q_{e}=\left[T_{e.in}\left(1-\frac{1}{\varepsilon^{m_{c}}}\right)\eta_{e}-\left(T_{h.in}-T_{d}\right)\left(1-\eta_{h}\right)\right]C_{p}$$
and
$$T_{ei}=T_{h.in}-\left(T_{h.in}-T_{d}\right)\eta_{h}$$
In addition to calculating the required air flow rate for air compression refrigeration, it is necessary to determine the cooling capacity of the water cooler. Considering the cooling capacity of the water cooler, we disregard the heat loss during the heat exchange process between the fluid and the water cooler. We assume that all the heat released by the air is completely carried away by the cooling water. Therefore, the heat exchange between the air and the cooling water can be expressed as follows:$$q=\dot{m}\cdot C_{p}\left(t_{in}-t_{out}\right)$$
The refrigeration efficiency of the system can be expressed as follows:$$COP=\frac{\dot{m}q_{e}}{P_{c.out}+q}$$

## 3. Modeling of Key System Components

The previous studies and calculations are based on conventional models of refrigeration systems. Whilst these models can determine the refrigeration performance of the system, they are not appropriate for a thorough examination of the system as the use of classical models is restricted.

To examine the system comprehensively, it is necessary to model each individual component in detail. In an air compression refrigeration cycle, the primary components include a water cooler, recooler, compressor, and expander. A turbine is commonly employed as an expander in refrigeration systems. As the working process of the expander is analogous to that of the compressor, we concentrate on modeling the former to preclude redundancy. Water cooler and recooler are both types of heat exchangers. However, the primary purpose of the water cooler is to decrease the temperature of high-pressure and high-temperature gas, thereby maintaining system stability, and the heat exchange of the water cooler should be sufficient. In contrast, the design of the recooler is instrumental in enhancing refrigeration system efficiency. Thus, we modeled the recooler heat exchanger for analysis.

### 3.1. Heat Exchanger Modeling

The heat exchanger stands as the primary locus of thermal exchange within the ambient wind tunnel of a high-speed locomotive, epitomizing its indispensability in modeling the air cooling system of the evaporative cooling mechanism. This article endeavors to precisely simulate the heat transfer dynamics of the heat exchanger by meticulously selecting the optimum number of effective heat transfer units while also delineating the resistive characteristics inherent in a plate heat exchanger.

Assuming a scenario wherein the fluid flows with impeccable conformity at the entrance of each flow channel within the heat exchanger, we further posit a symmetrical distribution of fluid temperature across the cross-sectional plane of every flow channel, disregarding the longitudinal conduction of heat within the interstitial bulkhead. Built upon this fundamental assumption, the heat transfer and resistance traits of each distinct set of flow channels in the heat exchanger remain uniform, permitting us to reduce the multidimensional flow within each channel to a unidirectional paradigm. In light of these considerations, we choose one of the aforementioned groups of flow channels for our comprehensive analysis.

Based on the law of conservation of energy, separate equations for the conservation of energy are established for the wall, hot fluid, and cold fluid, as follows:$$m_{w}c_{p,w}\frac{dT_{w}}{\mathrm{dt}}={\dot{Q}}_{h}-{\dot{Q}}_{c}$$
$${\dot{m}}_{h}c_{p,h}\frac{dT_{h}}{\mathrm{dt}}={\dot{m}}_{in,h}h_{in,h}-{\dot{m}}_{out,h}h_{out,h}-{\dot{Q}}_{h}$$
$$m_{c}c_{c,h}\frac{dT_{hc}}{\mathrm{dt}}={\dot{m}}_{in,c}h_{in,c}-{\dot{m}}_{out,c}h_{out,c}+{\dot{Q}}_{c}$$
When the heat exchanger operates in steady state, the actual heat exchange within the heat exchanger according to the ε−NTU method is given by
$${\dot{Q}}_{h}={\dot{Q}}_{c}=\varepsilon {\dot{Q}}_{\max}$$
where ${\dot{Q}}_{\max}$ is the theoretical maximum heat exchange.
$${\dot{Q}}_{\max}=\min\left(C_{p}{}_{{}_{{}_{h}}},C_{p}{}_{{}_{c}}\right)\left(T_{in,h}-T_{in,c}\right)$$
The heat capacities of the hot and cold fluids are denoted by ε.
$$\varepsilon =\frac{1-e^{-NTU(1-C^{*})}}{1-C^{*}e^{-NTU(1-C^{*})}}$$
$C^{*}$ is the heat capacity ratio when $C^{*}=1$.
$$\varepsilon =\frac{NTU}{1+NTU}$$
The number of heat transfer units (*NTU*) characterizes the heat transfer capacity of the heat exchanger and can be determined by the following equation:$$NTU=\frac{UA}{C_{\min}}$$
The product *UA* expresses the heat transfer per unit of average temperature difference within the heat exchanger. Based on the process of heat transfer within the heat exchanger, the calculation formula is as follows:$$\frac{1}{UA}=\frac{1}{{\left(\eta hA\right)}_{h}}+\frac{\gamma_{h}}{\eta A}+\frac{\delta_{w}}{\lambda_{w}A_{w}}+\frac{1}{{\left(\eta hA\right)}_{c}}+\frac{\gamma_{c}}{\eta A}$$
The above equation can be rewritten as
$$K=\frac{1}{a_{1}}+R_{1}+\frac{\delta_{w}}{\eta A}+R_{2}+\frac{1}{a_{1}}$$
$a_{1}$ and $a_{2}$ represent the heat transfer film coefficient.*R*_1_ and *R*_2_ represent the plate fouling factor.

The formula for the heat transfer film coefficient can be written as
$$a_{1}=Nu_{1}\frac{\lambda_{w}}{d_{w}}$$
where $Nu_{1}$ is a convective heat transfer quasi-count, the value of which is influenced by the flow state of the liquid in the system.
Nu1=CRe·Pr(0.3or0.4)
After obtaining the heat transfer coefficient of the heat exchanger and combining it with the air state at the inlet, the average temperature iteration can be employed to determine the outlet temperatures for both the hot and cold sides of the heat exchanger. The specific steps can be referred to in Figure 3.

The resistance of a fluid in a heat exchanger is generally calculated using Euler’s number.
Eu=bRed
The fluid resistance in the heat exchanger is mainly related to the fluid flow rate and the fluid density, the formula of which is as follows:Δp=bRedρω2=Euρω2

### 3.2. Expander Modeling

The turbine expander is an air compression refrigeration system that utilizes the output of cold air and shaft work components. By introducing compressed air through the inlet into the turbine for expansion, both external work is performed and the air temperature decreases as the pressure decreases. The gas undergoes an approximately isentropic adiabatic expansion process in the turbine, with the expansion ratio $\varepsilon_{t}$ being an important parameter of the expander, expressed as follows:$$\varepsilon_{t}=\frac{P_{in}}{P_{out}}$$
Assuming that the expansion process of the compressor’s expander is an approximately isentropic adiabatic compression multivariate process, the outlet temperature of the expander can be calculated using the following equation:$$T_{out}=T_{in}\left(1-\eta_{t}\left(1-\varepsilon_{t}{}^{\frac{1-k}{k}}\right)\right)$$
where $\eta_{t}$ represents the adiabatic efficiency of the expander, which expresses the ratio of the isentropic expansion output power of the expander to the actual output power.

Assuming that gas leakage occurring in the compressor is small and negligible, meaning that the mass flow rate of the imported and exported gas is conserved, the power required by the compressor can be calculated using the following equation:$$P_{c}=\dot{m}c_{p}\left(T_{in}-T_{out}\right)$$
During the previous system analysis, the aforementioned principles were followed in the modeling process. However, it was assumed that most of the expander turbine efficiency had a fixed value. In the air compression refrigeration cycle, the compressor turbine efficiency may vary significantly due to different operating conditions. Analyzing the efficiency of air compression systems with a fixed turbine efficiency can impact the accuracy of the results. To overcome this effect, a one-dimensional model of expander efficiency can be used to calculate the turbine efficiency, allowing for the utilization of dynamic turbine efficiency instead of a constant value.

The efficiency of a turbine expander generally consists of three components: peripheral loss, friction loss, and leakage loss. The overall efficiency of the expander can be expressed as
$$\eta_{tur}=1-\eta_{n}-\eta_{f}-\eta_{n}$$
In the formula, $\eta_{n}$ represents peripheral efficiency, $\eta_{f}$ represents friction efficiency, and $\eta_{l}$ represents leakage loss. The friction loss coefficient can be calculated as
$$\eta_{f}=f\frac{D_{1}^{2}}{v_{1}}{\left(\frac{u_{1}}{100}\right)}^{3}\frac{1}{1.36}\frac{1}{m_{f}\Delta h_{s}}$$
The leakage loss can be expressed as follows:$$\eta_{l}=\left\{\begin{array}{l} 1-1.3\frac{\delta }{l_{m}},0.015\leq \frac{\delta }{l_{2}}<0.05\\ 0.95-0.31\frac{\delta }{l_{m}},\frac{\delta }{l_{m}}\geq 0.05 \end{array}\right\}$$
The peripheral loss of the expander is a parameter that is highly influenced by the operating conditions, describing the various losses generated by the working mass during the peripheral work process. These include static lobe loss, dynamic lobe loss, and residual velocity loss. In a centripetal turbine, the mass flow is typically viscous, non-constant, and three-dimensionally complex. However, in an air compression refrigeration system, where air is the sole refrigeration medium, it can be simplified as a one-dimensional, axisymmetric, adiabatic, non-viscous, and stable flow. As a result, the efficiency of the centripetal turbine can be analyzed using a one-dimensional flow analysis method [14]. At the same time, to simplify the analysis process, we consider the flow of the work mass in the centripetal turbine as one-dimensional while ignoring the expansion process of the work mass in the worm shell and diffuser. Additionally, we assume that the gas velocity is below the speed of sound during the flow process.

The flow process of air from the compression refrigeration cycle into the working fluid in the turbine is illustrated in Figure 4. State 0 represents the initial state of the working fluid as it enters the turbine inlet radially. In the nozzle, high-pressure air expands from state 0 to state 1, resulting in a decrease in air enthalpy and an increase in flow rate. Subsequently, in the rotor, the organic vapor further expands from state 1 to state 2, leading to another decrease in air enthalpy and the conversion of original air energy into power output through the rotating blades. Finally, the exhausted organic vapor is discharged from the radial inflow turbine.

Figure 5 illustrates the velocity triangle of the expander during peripheral flow. The compressed air’s working process in the expander can be described by the following equation:$$c_{0}=\sqrt{2\cdot \Delta h_{s}}$$
Assuming *c*_0_ represents the ideal expansion velocity of the work mass in a centripetal turbine in m/s, Δ*h* represents the total enthalpy change during the operation of the expander.

For ease of characterization, the magnitude of the system’s velocity triangle is commonly expressed through the relative velocity:$${\bar{u}}_{1}=\frac{u_{1}}{c_{0}}$$
Thus, the peripheral efficiency of the expander can be expressed as
$$\eta_{u}=\frac{u_{1}\cdot c_{1u}-u_{2}\cdot c_{2u}}{\Delta h_{s}}=\frac{2\left(u_{1}\cdot c_{1u}-u_{2}\cdot c_{2u}\right)}{c_{0}^{2}}=2\left({\bar{u}}_{1}\cdot {\bar{c}}_{1u}-{\bar{u}}_{2}\cdot {\bar{c}}_{2u}\right)$$
Other variables in the equation can also be defined using parameters such as the ratio of circumferential efficiency to velocity, nozzle velocity coefficient, rotor blade velocity coefficient, reaction force, impeller diameter ratio, rotor inlet absolute velocity angle, and rotor outlet relative velocity angle in the peripheral design process.
$$\eta_{n}=2\varphi \cos\alpha \cdot x_{a}\sqrt{\left(1-\Omega \right)}-2{\bar{D}}^{2}x_{a}^{2}+2\psi \cos\beta \bar{D}x_{a}\sqrt{{\bar{D}}^{2}x_{a}^{2}+\Omega +\varphi^{2}\left(1-\Omega \right)-2\varphi \cos\alpha \cdot x_{a}\sqrt{\left(1-\Omega \right)}}$$
The speed ratio (*x_a_*) and the degree of reaction (Ω) are two crucial design parameters that significantly impact turbine efficiency. The speed ratio refers to the previously mentioned parameter. Furthermore, there exists a limiting relationship between various turbine parameters, such as the speed ratio and the degree of inversion [26].

By utilizing these design parameters, it becomes possible to determine additional design parameters in the expander system, such as the impeller inlet cross-sectional area and the rotor inlet diameter. The rotor inlet diameter can be expressed as
$$D_{l}=\sqrt{\frac{m_{0}}{\pi \left(l_{1}/D_{1}\right)\omega_{1}\cdot \sin\beta_{1}\cdot \rho_{1}\cdot \tau_{1}}}$$
where $\left(l_{1}/D_{1}\right)$ is the ratio of the height of the rotor blades to the diameter of the inner wheel, generally taken as 0.04.

In the determination of the relevant parameters, it is necessary to consider other design factors, such as the number of blades on the wind turbine. When optimizing these parameters, aiming for the simultaneous maximum peripheral efficiency of all parameters may lead to aerodynamic or structural design issues. Therefore, it is crucial to analyze and establish specific design parameters in advance. These include the static blade outlet Mach number, dynamic blade inlet impulse angle, and turbine rotational speed. By doing so, other design aspects can be appropriately constrained, ensuring the final selection of an aerodynamically, structurally, and reasonably sound scheme. This consideration guarantees the rationality of the chosen scheme in terms of aerodynamics, strength, and structure.

The literature reveals that the speed ratio (*x_a_*) and the reaction degree (Ω) of the centripetal turbine are the two design parameters that exert the greatest influence on the turbine’s efficiency, with the speed ratio mentioned previously. Furthermore, there exists a certain limiting relationship between each parameter of the turbine, such as the speed ratio and the degree of inversion (Ω) [28,29]. The speed ratio of the turbine stands as the most critical design parameter affecting levelling efficiency. It is influenced by both turbine-related design parameters and the turbine’s operating state. Consequently, in the actual modeling process, the optimal speed ratio for turbine operation is generally determined based on the import and export parameters of the turbine. This determination can be achieved using the above-mentioned formula and the circumferential efficiency, yielding the following magnitude:$$\begin{array}{l} m=\frac{1}{\psi^{2}}\left(1-\sqrt{\frac{{\bar{D}}^{2}\left(1-\varphi^{2}\right)\left(1-\cos^{2}\beta \right)\psi^{2}}{\varphi^{2}\cos^{2}\alpha +{\bar{D}}^{2}\left(1-\varphi^{2}\right)}}\right)\\ x_{a.opt}=\frac{\psi }{\sqrt{{\bar{D}}^{2}\left(\frac{\cos^{2}\beta }{m^{2}}-\psi^{2}\right)+\frac{\left(1-m^{2}\psi^{4}\right)\varphi^{2}\cos^{2}\alpha }{m\psi^{2}\left(1-\varphi^{2}\right)}}} \end{array}$$
When modeling the expander, it is only necessary to select different degrees of inversion. However, since numerous parameters have already been determined, thereby limiting the variation in the overall size of the turbine, determining the performance of compressors of different sizes requires applying the principle of similarity to vary all parameters proportionally.

Similarity theory, initially applied in geometry, refers to situations where two figures possess similar geometric shapes but differ in geometric dimensions. In the context of turbines, similarity theory is primarily employed in two aspects. Firstly, during the design of a new turbine, the performance curve of the new machine is deduced from an existing turbine model. Secondly, when testing the machine, certain test conditions such as inlet conditions, mass, and speed might not align with the design conditions. In such cases, the aerodynamic performance of the machine must be correctly converted from the test conditions to the design conditions. The application of similarity principles relies on satisfying four similarity conditions: the deterministic criterion for characterizing viscous effects; the deterministic similarity criterion for characterizing compressibility; the deterministic similarity criterion for non-constant flow; meeting geometrical similarity;similarity of inlet velocity graphs; equality of the Mach number and Reynolds number;and equality of the gas constant entropy exponent k.

To compare the size scale between different turbines, we introduce the gauge coefficients, which is defined as follows:$$\frac{D_{2}}{D_{1}}=m_{l},\;\frac{A_{2}}{A_{1}}=m_{l}{}^{2}$$
When the constant entropy indices of the two turbines are equal, denoted as $k_{1}=k_{2}$, the condition ensuring the equality of the corresponding Mach numbers in the system can be expressed as follows:$$\frac{u_{1}}{{\sqrt{RT}}_{in.1}}=\frac{u_{2}}{{\sqrt{RT}}_{in.2}}$$
$$n_{2}=\frac{1}{m_{l}}\sqrt{\frac{RT_{in.1}}{RT_{in.2}}}n_{1}$$
If the inlet conditions of the two turbines are the same, the relationship between the corresponding rotational speeds of the turbines is solely dependent on their relative sizes. Hence, during the process of model simulation, a larger turbine will have a lower rotational speed compared to a smaller turbine.

In the case of turbines, the constant entropy energy head of the system is related by the following equation:$$h_{s}=W_{s}=\frac{k}{k-1}RT_{in}\left(\varepsilon^{\frac{k-1}{k}}-1\right)$$
The turbine compression ratio, denoted as ε, can be derived from the gas continuity equation. According to the similarity principle, two turbines that comply with this principle will have equal compression ratios, $\varepsilon =\varepsilon^{\prime }$. Additionally, since the gas constant entropy index (*k*) is also equal for two similar turbines, it follows that
$$\frac{W_{s1}}{RT_{in1}}=\frac{W_{s2}}{RT_{in2}}$$
The system energy head is determined by
$$\psi_{s1}=\frac{h_{s1}}{u_{1}^{2}},\;\psi_{s2}=\frac{h_{s2}}{u_{2}^{2}}$$
According to the principle of similarity, the Mach numbers of the two turbines are equal, resulting in
$$Ma_{1}=Ma_{2},\;\mathrm{and}\;\frac{u_{1}}{\sqrt{RT_{in1}}}=\frac{u_{2}}{\sqrt{RT_{in2}}}$$
By substituting the previous equation, it can be shown that the energy heads of similar turbine systems are equal.
$$\psi_{s1}=\psi_{s2}$$
The equation for the volume flow rate of the system is given by
$$q_{v1}=c_{1}A_{1}$$
This can be derived by analyzing the gas velocity across its passage during operation.
$$\frac{c_{1}}{u_{1}}=\frac{c_{2}}{u_{2}}$$
It is also known that the formula for the speed of the turbine is
$$n_{1}=\frac{60u_{1}}{\pi D_{1}},\;n_{2}=\frac{60u_{2}}{\pi D_{2}}$$
Therefore, it can be inferred that
$$q_{v2}=m_{1}{}^{3}\frac{n_{2}}{n_{1}}q_{v1}$$
The mass flow rate is related to
$$q_{m1}=q_{v1}\rho_{in1}=q_{v1}\frac{p_{in1}}{RT_{in1}}$$
$$q_{m2}=q_{v2}\rho_{in2}=q_{v2}\frac{p_{in2}}{RT_{in2}}$$
From this, we can infer that
$$q_{m1}\frac{\sqrt{RT_{in1}}}{p_{in1}}=q_{m2}\frac{1}{m_{l}{}^{2}}\frac{\sqrt{RT_{in2}}}{p_{in2}}$$
Moreover, since the turbine power is the product of the energy head and the mass flow rate,
$$P_{1}=W_{tot1}q_{m1},\;P_{2}=W_{tot2}q_{m2}$$
Hence,
$$\frac{P_{1}}{W_{tot1}q_{m1}}=\frac{P_{2}}{W_{tot2}q_{m2}}$$
Further simplification yields
$$\frac{P_{1}}{p_{in1}\sqrt{RT_{in1}}}=\frac{1}{m_{l}{}^{2}}\frac{P_{2}}{p_{in2}\sqrt{RT_{in1}}}$$
It can be observed that the changes in mass flow rate and power for both turbines are proportional to the square of ml. Using ml to express the change in turbine performance may not be immediately clear; therefore, a change in the rated mass flow rate is sometimes used instead. As achieving strict geometric similarity between turbines is challenging, biases may arise in this conversion, particularly when the turbine sizes of the two units differ significantly. To address this problem, the relevant literature suggests parameter corrections and approximations.

A one-dimensional analysis model and a turbine similarity principle code have been developed in-house by the authors using Simulink in MATLAB 2018b. The calculation procedure is illustrated in Figure 6. The initial parameters [17] are also indicated, and the geometry size of the radial inflow turbine and the velocity triangle at each characteristic section can be calculated

Xia [13] conducted a one-dimensional flow analysis and created a three-dimensional model. This article presents a plot of the mass flow rate versus the turbine efficiency in Figure 7 when air is utilized as the fluid. The main turbine parameter specified in the article is a reaction degree of 0.51 and a blade length of 54.3 mm. In Li’s article [25], a turbine with a reaction degree of 0.51 and a blade length of 8 was simulated, and relevant experiments were carried out. We can observe from Figure 8 that the results of our simulation method and the original simulation method are essentially identical through comparison.

## 4. Results and Discussion

This section comprises two main parts. Firstly, we conduct a thermodynamic analysis of the environmental wind tunnel refrigeration system designed for high-speed trains, which utilizes an air compression refrigeration system to ascertain its thermodynamic performance, and we compare our design to a conventional evaporative refrigeration system. Secondly, a sensitivity analysis was conducted for the key parameters in the refrigeration system using the established model. The main methodology involved determining the impact of different parameters on system performance via simulation numerical modeling.

### 4.1. Comparison of Air Compression Refrigeration Cycle and Evaporative Refrigeration Cycle

The impact of the air cooling cycle on the cooling efficiency of air compression refrigeration systems was studied under the assumption of demand conditions that are identical to those of evaporative cooling. In the experimental section of the train, the temperature was maintained at −70 °C, and the wind speed was set at 100 m/s. Previous field simulations indicated that the required cooling capacity for the application of air compression refrigeration is approximately 1.8 × 10^7^ W. This is because during the actual working process, a certain amount of air needs to be extracted from the wind tunnel for the air compression refrigeration cycle. As a result, the airflow in the second half of the wind tunnel decreases since the real air flow becomes smaller. Due to the constant air density, this reduction in airflow leads to a decrease in the air velocity within the wind tunnel, thereby reducing flow losses. Consequently, the actual amount of cooling required to achieve the same experimental conditions (100 m/s wind speed and −70 °C) in the experimental section is lower compared to evaporative cooling.

In addition to accounting for the cooling requirements of the wind tunnel, the quality of the external air also impacts the efficiency of the cooling cycle. Air compression cooling necessitates the compression of external air to obtain a cooling source. For the analysis, dry air with a temperature of 25 °C and a pressure of 1.0 × 10^5^ Pa was selected. Other main equipment parameters include an inlet compression ratio of 2.5, a cooling water temperature of 40 °C for the water cooler, and a source gas temperature of −40 °C for the return cooler. The return cooler is connected to the ambient wind tunnel and possesses a sufficiently large heat transfer area to ensure the adequate cooling of the compressed air. This process allows the final compressed air to exit the return cooler at the same temperature as the gas drawn from the wind tunnel.

The mechanical energy generated by the turbine expander, through the work of air expansion, is fully utilized in the turbo compressor air compression system. The pressure–enthalpy and temperature–entropy diagrams for the system are shown in Figure 9.

The operating conditions of the different components in the different air cooling systems were also obtained in Table 1.

At this stage, the air compression system energy flow diagram is shown in Figure 10, and an external energy input of approximately 28.3 MW is required to supply the refrigeration system, considering the cooling efficiency of the water-cooling tower as 4.5. This yields a coefficient of performance (COP) for the system of approximately 0.636, and the energy flow diagram of the refrigeration system is shown below.

To compare the cooling effect of the air compression system, a conventional cascade refrigerating system was introduced, and the system schematic is shown in Figure 11.

Assuming that the temperature in the experimental section of the train is still −70 °C and the wind speed is 100 m/s, the power loss of the system is reduced due to the air extraction required for air compression cooling. Additionally, the resistance in the wind tunnel increases as a result of the additional heat exchange network. Therefore, under the same conditions in the experimental section, the evaporative cooling cycle method needs to provide more cooling capacity in order to achieve the desired experimental environment. Simulation calculations indicate that the refrigeration system needs to deliver a cooling capacity of approximately 2.0 × 10^7^ W to achieve the same experimental conditions in the wind tunnel section.

With an external ambient temperature of 25 °C, an evaporator temperature of −70 °C, a condensing temperature of 40 °C, and an evaporator–condenser temperature difference of 5 °C, the midpoint temperature of the evaporative refrigeration cycle was determined to be −23 °C. By observing the characteristics, R23 was selected as the refrigerant for the low-temperature part of the refrigeration system, and R404A was chosen for the high-temperature part, allowing for the design of the experimental state at each point. Calculations are carried out using Solkane, and the completion of pressure enthalpy and temperature entropy diagrams are shown on Figure 12.

The operating conditions of the various components in different evaporative refrigeration systems were obtained, as shown in Table 2.

At this point, an external energy input of approximately 32.1 MW is required to supply the refrigeration system (assuming a cooling efficiency of 4.5 for the water-cooled tower), resulting in an approximate system cooling efficiency COP of 0.624.

The comparison reveals that, although the cooling efficiency of air compression refrigeration is not significantly different from evaporative refrigeration, the air compression system has a lesser impact on the cooling section of the wind tunnel. Consequently, it requires less energy for the refrigeration system under the same experimental conditions. Specifically, the air compression system necessitates approximately 3.72 MW less energy, resulting in an improvement of approximately 13.15% over the efficiency of the evaporative refrigeration system.

### 4.2. Systematic Sensitivity Analysis

A sensitivity analysis is conducted for the design parameters of the aforementioned components, encompassing two main areas. The first area involves studying the influence of various parameters of the refrigeration equipment on the system under specific external conditions. The second area explores the impact of different external conditions on the cooling sensitivity of the system once the refrigeration equipment has been selected.

Heat exchanger:

The initial step is to assess the effect of different design parameters on the heat transfer capacity of the system’s return cooler and the cooling efficiency of the air compression refrigeration system. For this particular case, a plate heat exchanger was chosen as the return cooler. Design parameters for the plate heat exchanger include total heat transfer area, single plate heat transfer area, plate configuration form, plate spacing, plate thickness, individual flow path cross-section, etc. While these design parameters are somewhat inter-related, the total heat transfer area of the heat exchanger exhibits the greatest relative impact on heat transfer performance. It serves as an essential indicator for evaluating the heat transfer capacity of the plate heat exchanger and can be derived from the product of the heat transfer area per single plate and the number of heat exchanger plates.

Assuming that the other design parameters of the return cooler remain constant, and with inlet temperatures of 313 K and 228 K, respectively, for the hot and refrigeration sides of the return cooler, as well as inlet pressures of 350 Kpa and 110 Kpa, the sole parameter adjusted is the total heat transfer area of the heat exchanger (achieved by altering the number of heat exchanger plates). The relationship between the heat exchanger area and the outlet temperature of the return cooler is presented in Figure 13a below, indicating a considerable decrease in the outlet temperature as the system’s flow rate changes. Moreover, the outlet temperature reduces at a faster rate when the flow rate is smaller. When the temperature drops to a certain level, the capacity for heat transfer of the heat exchanger reaches its limit. Increasing the heat transfer area will not lead to a reduction in outlet temperature. When the flow rate of the heat exchanger remains constant, adjusting the inlet temperature of the cooler return, heat exchanger heat transfer area, and export temperature are shown in Figure 13b below. It can be observed from the graph that the increase in heat transfer area results in a decrease in the heat exchanger export temperature. Similarly, the graph reveals that the outlet temperature of the heat exchanger drops faster with an increase in the inlet temperature of the heat exchanger. It is observable from Figure 13c that, as the heat transfer area increases, the outlet pressure of the heat exchanger decreases. Furthermore, with an increase in flow rate through the heat exchanger, the decrease in outlet pressure becomes more pronounced. However, the overall decrease in pressure is insignificant when compared to the inlet pressure of the system.

Further analysis of the effect of changes in the heat transfer area of the return cooler on the cooling efficiency of the air compression refrigeration system assumes that the system is as described above, with unchanged parameters for the components. In this system, the compressor increases the atmospheric gas to 250 Kpa and subsequently recovers it to 110 Kpa through a turbine–compression system. The high-pressure gas then passes through a water cooler, reducing its temperature to 45 °C. A certain amount of gas is extracted through the wind tunnel and used as the cold source gas in the recooler, where the temperature of the cold source gas is −45 °C. It should be noted that if the system efficiency is below 0.2, the air compression refrigeration system will not function effectively. With variations in the heat exchanger area, the system experiences changes in heat transfer efficiency, as shown in Figure 14. It can be observed that an increase in heat transfer area leads to a rapid improvement in the system’s heat transfer efficiency. Additionally, it can be seen that a higher mass flow rate in the system results in a slower increase in cooling efficiency. Furthermore, by examining the pattern of heat exchanger outlet temperature and pressure in the graph above, it becomes evident that this is due to the rapid decrease in outlet temperature as the heat transfer area increases. As the outlet temperature of the return cooler decreases, the temperature of the refrigerant gas is lower after expansion, allowing for more cooling capacity per unit mass flow rate, thereby increasing the refrigeration efficiency of the system.

However, once the heat transfer efficiency reaches its peak value, any further increase in heat transfer area results in a gradual decline in system cooling efficiency. Moreover, this decline is more pronounced with higher mass flow rates in the system. This phenomenon occurs because after reaching a certain limit of heat transfer efficiency, further increases in heat transfer area no longer reduce the outlet temperature of the system. Instead, they increase the resistance within the heat exchanger, leading to a drop in outlet pressure within the system. Consequently, the subsequent expander work is reduced, resulting in a decrease in air temperature and an increase in the refrigeration efficiency of the system.

It can be concluded that for the air compression system, the heat transfer area of the return cooler should be appropriately increased to ensure a lower exit temperature. This is essential as reducing the exit temperature effectively improves the refrigeration efficiency of the system. Although excessive heat transfer area may have a certain impact on system efficiency, it is smaller compared to other factors. Therefore, during design selection, it is advisable to choose a heat exchanger with a larger heat transfer area as the system’s return cooler.

During the operation of the environmental wind tunnel for high-speed trains, various operating conditions arise, necessitating the study of unified heat exchanger performance at different mass flow rates. Assuming the system’s design parameters remain constant, the hot side inlet temperature and pressure of the return cooler are 313 K and 350 Kpa, respectively, while the cooling side’s inlet temperature and pressure are 228 K and 110 Kpa, respectively. Only the heat exchanger’s mass flow rate is varied to observe how the outlet temperature and pressure change with different heat transfer areas. As depicted in Figure 15a, as the system’s mass flow rate increases, the heat exchanger’s outlet temperature rises initially, and then reaches a peak before declining. This trend becomes more pronounced when the heat exchanger area is smaller. When the heat transfer area is sufficiently large and the overall heat transfer rate within the system is low, the system consistently operates near its cooling limit. The relationship between the heat exchanger’s mass flow rate and its outlet pressure is demonstrated in the following Figure 15b. It can be observed that increasing the fluid’s mass flow rate through the heat exchanger leads to a decrease in outlet pressure. Moreover, as the total heat transfer area within the heat exchanger expands, the outlet pressure reduction becomes more significant. However, this overall decline in pressure remains relatively small compared to the system’s inlet pressure.

Further analysis of the effect of using different heat transfer areas on the cooling efficiency of air compression refrigeration systems with variations in mass flow rates is presented. This study assumes that the remaining components of the system operate at constant efficiency. In this system, the compressor increases the atmospheric gas pressure to 250 KPa, and then the turbine–compressor component restores the system pressure to 110 Kpa. The high-pressure gas passes through a water cooler, reducing its temperature to 45 °C. A certain amount of gas is extracted through the wind tunnel and used as the source gas in the recooler, where the temperature of the source gas is −45 °C. It is assumed that when the system efficiency falls below 0.2, the air compression refrigeration system cannot function effectively.

With changes in the heat exchanger area, the system’s heat transfer efficiency varies, as shown in Figure 16a,b. When the heat exchanger’s working mass flow rate is low, the system’s refrigeration efficiency remains mostly constant. Additionally, the heat transfer efficiency of the smaller heat exchanger is slightly higher than that of the larger heat exchanger. However, when the mass flow rate of air in the system reaches a certain point, the efficiency of the smaller heat exchanger starts to rapidly decrease, while the larger heat exchanger continues to decrease slowly. This trend can be attributed to the fact that at lower mass flow rates, the outlet temperature of the system cooler remains within a low range. Hence, the main factor influencing the system’s efficiency becomes the change in outlet pressure. However, as the mass flow rate in the system increases, the heat transfer area of the system cooler fails to maintain the outlet temperature within the lower range. Consequently, the outlet temperature of the system cooler rises rapidly, leading to a decline in the system’s overall efficiency. Simultaneously, the outlet pressure in the system continues to gradually decrease. The combined effect of these factors results in a rapid decrease in the cooling efficiency of the system.

While the cooling efficiency of the system is an important parameter, more attention is typically given to the cooling capacity of the air compression system. The cooling capacity is determined by the difference between the inlet and outlet temperatures and the mass flow rate of air in the system. If the mass flow rate of the system changes significantly while the two parameters of the system gas remain largely unchanged, the cooling capacity can be affected. In this study, it was observed that an increase in the mass flow rate of the system leads to an increase in the outlet temperature of the return cooler, subsequently raising the temperature of the system’s cold air. As a result, the cooling efficiency of the system decreases, directly affecting its cooling capacity. Moreover, when the mass flow rate exceeds a certain range, the refrigeration efficiency of the air compression system falls below 0.2, rendering the entire refrigeration system incapable of delivering a qualified cooling source for the high-speed train’s environmental wind tunnel. When the mass flow rate in the system is low, the cooling capacity increases proportionally with the mass flow rate, and the growth rate remains relatively consistent. The heat exchanger with a larger heat transfer area exhibits only slightly lower efficiency than the heat exchanger with a smaller heat transfer area. However, as the mass flow rate increases, it becomes evident that insufficient heat exchange area in the system causes the cooling capacity to decrease instead of increasing.

From the above analysis, it can be concluded that the heat transfer area of the return cooler has a significant impact on both the cooling efficiency and cooling capacity of the system. However, this impact is mainly notable when the heat transfer area of the return cooler fails to meet the system’s mass flow rate requirements. At this point, an increase in the mass flow rate leads to a rapid decline in the system’s cooling efficiency, while the cooling capacity remains unchanged. Conversely, when the heat transfer area of the return cooler is sufficiently large, the system’s refrigeration efficiency only experiences a gradual decline. Therefore, during system design and selection, it is advisable to choose a heat exchanger with a slightly larger heat transfer area based on the system’s requirements. This decision facilitates the improvement of the system’s efficiency. Although increasing the heat transfer area may raise the construction cost to some extent, as indicated by the previous analysis, the overall impact on the system’s economy is insignificant compared to operational and maintenance costs. Hence, it can be assumed that when the heat transfer area of the recuperator in the system is adequately large, any subsequent changes in heat transfer area or mass flow rate will have minimal impact on the system’s economic performance. Consequently, the change in heat transfer area can be considered to have a low sensitivity.

2.Turbine Expander

The sensitivity of turbine expander design parameters has been extensively studied in previous articles [30,31]. These studies employed the extreme difference method to analyze the impact of different parameters on expander efficiency. In a practical expander design, it is often challenging to optimize every parameter due to considerations of aerodynamic characteristics and design feasibility. Therefore, this paper focuses on investigating expander inversion as the main design parameter. However, since expander inversion is a dimensionless number, it cannot solely determine the values of other expander design parameters. For this reason, we refer to the expander design parameter selection principles described by Li Peng [28] as a guideline.

In our study, an expander with an applicable mass flow rate of approximately 360 kg/s and a design inversion ranging from 0.3 to 0.8 was selected. It should be noted that the original expander might not meet the requirements of the wind tunnel for high-speed train environments. Hence, based on similar principles, three times the amplification parameters were set and used to select the final compressor. We acknowledge that using multiple compressors in parallel could achieve the same effect, but we do not investigate this approach in this paper.

Figure 17a,b depicts the efficiency and outlet temperature of the expander for different mass flow rates, showing that the expander achieves maximum efficiency at approximately 0.51. As inversion increases, expander efficiency gradually improves until it reaches the optimum value, after which it rapidly declines. The graphs also reveal that changes in mass flow rate have a limited impact on the overall trend of expander efficiency, but they do affect the maximum efficiency value. Similarly, the outlet temperature of the expander follows a similar pattern to its efficiency, initially decreasing with increasing inversion and then rapidly rising after reaching a minimum temperature. In the context of an air compression refrigeration system, assuming the system follows the description mentioned above with constant efficiency for the remaining components, the compressor is capable of increasing atmospheric gas pressure to 250 KPa. By employing a booster compressor, the system further pressurizes the gas and utilizes the expander to return the pressure to 110 KPa. The high-pressure air’s outlet temperature in the water cooler is maintained at 45 °C, and the return cooler features a heat transfer area of 800 m^2^, with an outlet temperature consistently maintained at −40 °C. The outlet pressure follows the aforementioned model law. The expander’s efficiency under these conditions is illustrated in Figure 17c. This behavior can be attributed to the fact that the expander converts the internal energy of high-pressure air into mechanical energy during system operation, which drives the booster compressor. Consequently, the compressed air pressure decreases, leading to lower enthalpy within the system. As the high-pressure gas enters the expander, the reduced work capacity of the air causes the outlet air temperature to rise, resulting in a reduced cooling capacity for the system.

The changes in turbine performance, system performance, and cooling capacity at different air mass flow rates, while keeping the degree of reactivity constant, are further investigated. The remaining settings of the turbine model remain the same as before. The simulation results are shown in Figure 18a–c. When the turbine operates at an air mass flow rate lower than its design mass flow rate, the turbine efficiency increases as the mass flow rate increases. The turbine achieves its highest efficiency when the mass flow rate matches the design mass flow rate, resulting in the lowest turbine outlet temperature and the highest external work capacity. However, when the mass flow rate exceeds the design mass flow rate, the turbine can still function, but its operating efficiency decreases. Additionally, the outlet temperature of the turbine increases, reducing the system’s cooling capacity. Nevertheless, although the turbine’s efficiency decreases, more air passes through the turbine, resulting in increased external work capacity. It is worth noting that the decrease in the system’s efficiency due to the increase in air mass flow rate has a greater impact than the decrease in turbine efficiency. Consequently, the external power output of the turbine decreases, but at a slightly slower rate than the decrease in system efficiency.

The performance of air compression refrigeration systems follows similar to turbine operation, with an initial increase in efficiency followed by a decrease. However, the rate of change in the system is much greater than the rate of change in the turbine itself, as explained in the previous section. Comparing Figure 19a,b reveals that the mass flow rate at which the system operates at maximum efficiency differs from the mass flow rate at which the system produces maximum cooling capacity. The latter value is slightly greater than the former because the cooling efficiency characterizes the cooling capacity of the air per unit mass. As the air mass flow rate increases, the total cooling capacity of the system also increases, despite a decrease in the cooling capacity per unit mass of air.

In the daily selection process, it is necessary to match the equipment’s scale factor with the operating mass flow rate so that the turbine system can operate at the appropriate mass flow rate. An expander with a rated flow rate of 360 kg/s was selected as a benchmark. In order to study the impact of different magnitude coefficients on the expander’s performance, 330 kg/s and 390 kg/s expanders were built using similar principles for comparison. Figure 20a illustrate how the turbine operating efficiency varies with the turbine mass flow rate for different scale factors. The graph demonstrates that all turbines have the same maximum efficiency point of approximately 82.14% due to identical turbine design parameters. Furthermore, the turbines exhibit a consistent trend of first increasing and then decreasing. Although the rate of change in turbine efficiency differs slightly across various scale factors, larger scale factor turbines generally experience a slightly lower rate of change compared to smaller turbines. However, it is important to note that turbines with different scale factors are not suitable for the same operating range, and the corresponding optimum mass flow rate for turbine operation varies. The turbine outlet temperature and efficiency display a similar trend, as shown in Figure 20b, albeit in opposite directions. Additionally, the external work capacity of the turbine is a crucial aspect of interest as it directly impacts the cooling efficiency of the system. Figure 20c depicts the external work power of a turbine for different scale factors plotted against the mass flow rate of the compressed gas. This graph reveals that the external power output of the turbine increases with the mass flow rate and begins to decline rapidly after reaching its peak. Moreover, the magnitude of the turbine’s external power output is influenced by its scale factor, with higher scale factors resulting in greater turbine external power and corresponding mass flow rates of compressed air.

For the entire air compression refrigeration system, Figure 21a show the variation in air mass flow rate versus refrigeration efficiency. As depicted, the change in system efficiency remains essentially the same across different scale factors, except that larger turbine magnitude factors require a higher mass flow rate to achieve optimum system efficiency. The maximum cooling efficiency remains constant Figure 21b, irrespective of scale factors. Furthermore, Figure 20b illustrates the change in system cooling capacity with system air mass flow rate for different magnitude factors. A comparison reveals that although the maximum cooling capacity of the system differs, the pattern of change is similar to that of the system’s cooling efficiency, initially decreasing and then increasing. It is also evident that when the air mass flow rate of the system is small, a smaller turbine size may result in a larger cooling capacity. This phenomenon occurs because operating the system outside the suitable mass flow rate range significantly reduces its cooling efficiency, thus affecting the overall cooling capacity. Therefore, selecting a turbine with an appropriate scale factor that matches the air mass flow rate is crucial for enhancing system performance.

Regarding the relationship between the turbine’s gauge coefficients (rated flow) and the maximum cooling capacity of the system, Figure 22a,b illustrates that the cooling efficiency of the system slightly decreases with increasing turbine gauge coefficient and the drop in pressure of the recooler outlet may have caused this reduction. However, as the mass flow rate of air in the system increases with the turbine gauge coefficient, the maximum cooling capacity of the system also rapidly increases (Figure 22b). Therefore, when selecting an air compression refrigeration system, particular attention should be paid to the actual size of the turbine to ensure it meets the maximum cooling capacity requirements of the system.

This analysis demonstrates that the scale factor and degree of inversion of the turbine have a significant influence on the system’s operating efficiency and cooling capacity. These design parameters are highly sensitive in air compression refrigeration systems and should be considered as important optimization parameters in subsequent parameter optimization processes. Additionally, since the compressor and turbine belong to the same rotating machinery and operate based on similar principles, the design parameters of the compressor in the air compression refrigeration system are equally important. The compressor plays a crucial role in increasing the gas pressure and is considered to be no less significant than the turbine in the system. Thus, the design parameters of the compressor should be an important target for subsequent optimization.

Based on these results, the design parameters identified as highly sensitive in air compression refrigeration systems include the inverse motion and magnitude coefficients of the compressor and turbine. Conversely, the system exhibits relatively low sensitivity parameters for the design parameters of the heat exchanger.

## 5. Conclusions

This paper proposes a novel cooling solution for high-speed train wind tunnels as an alternative to previous methods. The proposed solution utilizes an air compression cycle based on the Brayton cycle. It generates dry air at ambient temperature and pressure, which is then injected directly into the wind tunnel to serve as a cooling source instead of employing traditional evaporative cooling methods. Due to the complexity of the new system, dynamic modeling of the refrigeration equipment and sensitivity analysis of key design parameters were conducted to facilitate subsequent equipment selection optimization. The key findings are as follows:Through the evaluation of two refrigeration cycles, it was found that the evaporative refrigeration cycle has a refrigeration efficiency of about 0.624 and requires approximately 32.1 MW of external energy for operation. On the other hand, the air compression refrigeration system has a COP refrigeration efficiency of about 0.636 and needs 28.3 MW of energy to function. The comparison confirms the air compression system’s superiority in energy efficiency, requiring 3.72 MW less energy from external sources compared to the evaporative refrigeration system. The air compression system consumes 3.72 MW less external energy to operate compared to the evaporative refrigeration system, with a 13.15% improvement in efficiency.A dynamic model of the heat exchanger was developed using the heat transfer effectiveness approach. This model enables the calculation of heat transfer efficiency based on outlet temperature, pressure, and the design dimensions and operating parameters of the heat exchanger. The modeled heat exchanger was employed as a return heat exchanger in the refrigeration system. The study revealed that an increase in the heat transfer area of the return cooler led to a decrease in outlet temperature while maintaining a constant mass flow rate. However, beyond a certain threshold, the temperature of the heat exchanger remained largely unchanged. Additionally, the outlet pressure of the recuperator gradually decreased as both the heat transfer area and mass flow rate of air passing through it increased.One-dimensional flow characteristics analysis and the similarity principle were applied in the modeling of rotating machinery, such as turbines and compressors. The operational efficiency of these machines was initially established by analyzing the one-dimensional flow of the model. This involved utilizing velocity triangle and energy loss models to study turbine operation under different dimensional and operational parameters. To ensure rational aerodynamic and structural changes during the system optimization process, the concept of dimensional coefficients was introduced based on the principle of similarity. The study of turbine parameters focused on the turbine reaction degree and magnitude coefficient as key design optimization indicators. It was found that optimal values for both the turbine reaction degree and magnitude coefficient existed under specific operating conditions.System sensitivity analysis was conducted for various design parameters. The influence of the heat transfer area of the return cooler on the cooling efficiency and capacity of the system was only significant when it failed to meet the heat transfer requirements of the system. Once the heat transfer area reached a sufficient size, further increases had a limited impact on the cooling efficiency. In the case of turbines and compressors, variations in the reactance and magnitude coefficients of the equipment had a substantial impact on the cooling efficiency and capacity of the system. Moreover, there were optimal reactance and magnitude coefficients for different cooling demands, making them highly sensitive design parameters that should be prioritized in subsequent optimization studies.

## Figures and Tables

**Figure 1 entropy-25-01312-f001:**
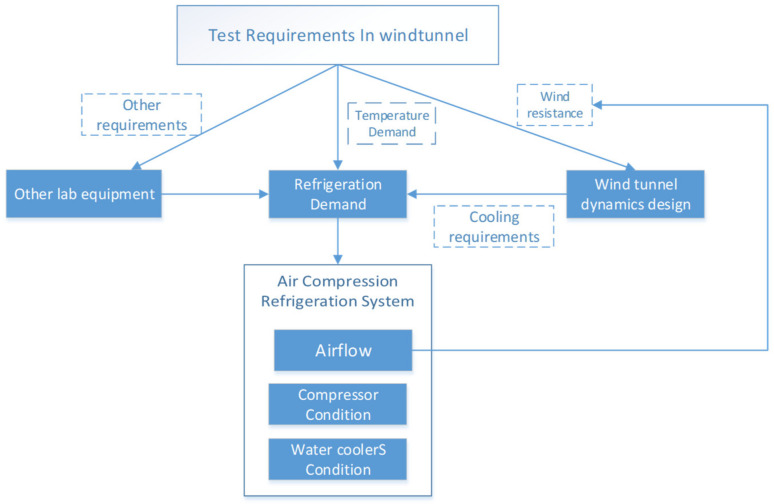
The cooling requirements of the wind tunnel.

**Figure 2 entropy-25-01312-f002:**
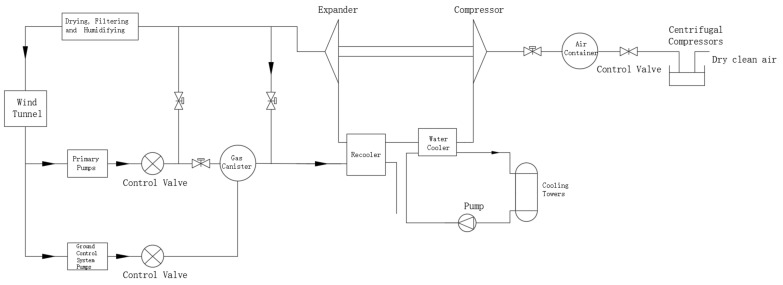
Air compression refrigeration system.

**Figure 3 entropy-25-01312-f003:**
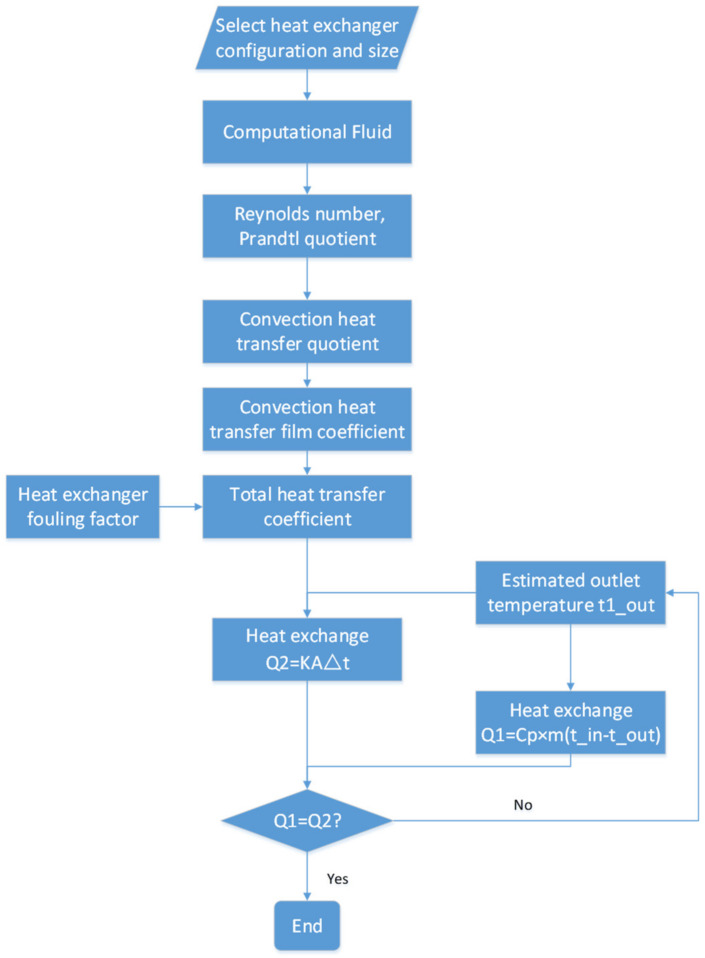
Heat exchanger modeling flowchart.

**Figure 4 entropy-25-01312-f004:**
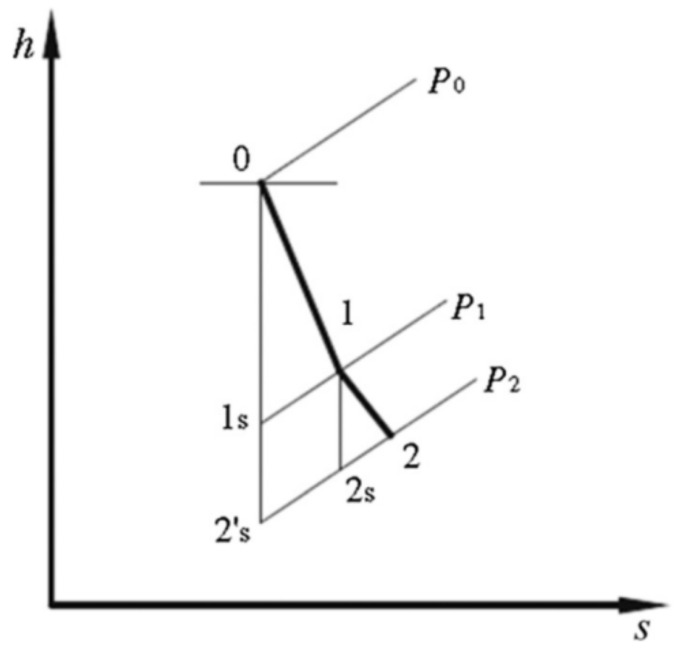
Theh-s diagram of the working fluid in the radial inflow turbine.

**Figure 5 entropy-25-01312-f005:**
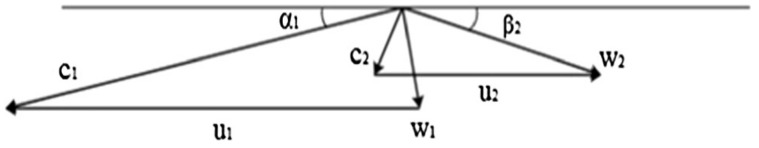
Velocity triangles of the radial inflow turbine.

**Figure 6 entropy-25-01312-f006:**
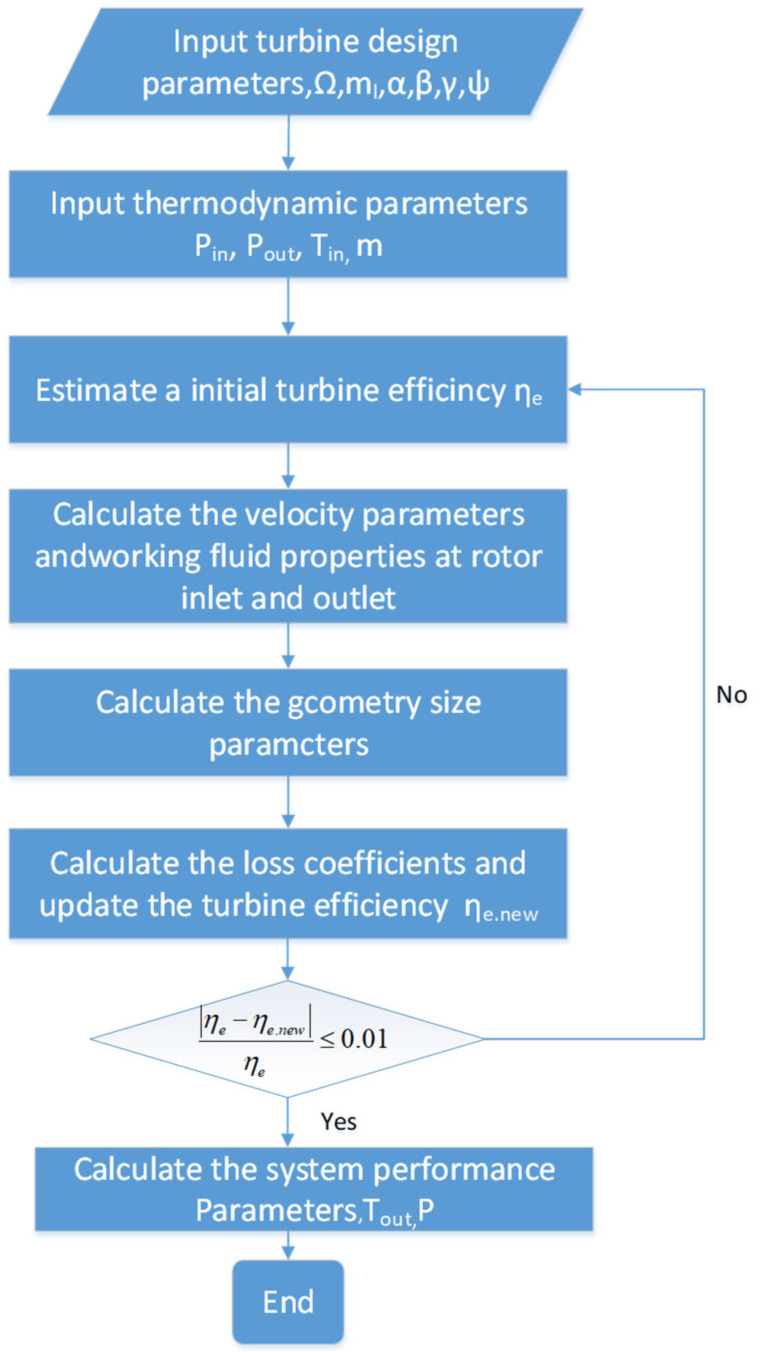
Turbine modeling flowchart.

**Figure 7 entropy-25-01312-f007:**
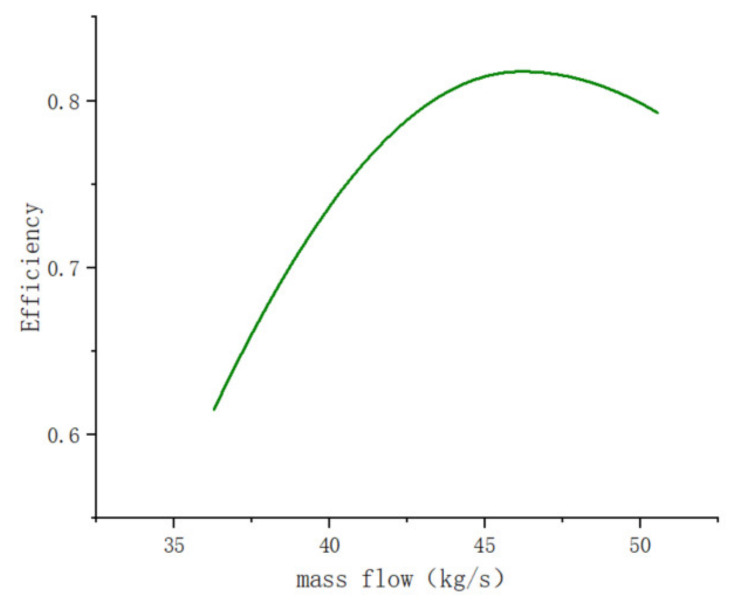
Turbine expander mass flow and efficiency.

**Figure 8 entropy-25-01312-f008:**
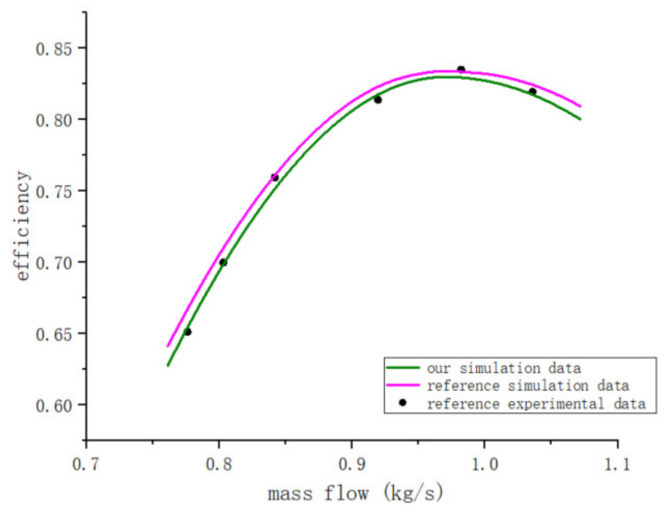
Comparison of the results of our code and the reference.

**Figure 9 entropy-25-01312-f009:**
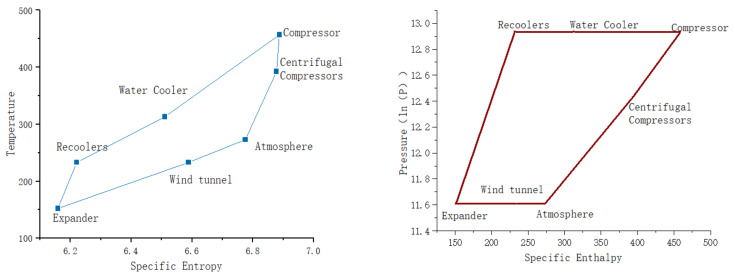
T-s and P h diagrams for the air compression refrigeration cycle.

**Figure 10 entropy-25-01312-f010:**
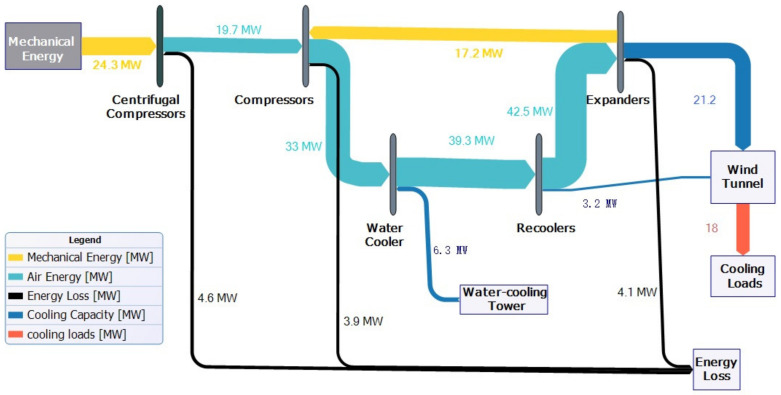
Air compression system energy flow diagram.

**Figure 11 entropy-25-01312-f011:**
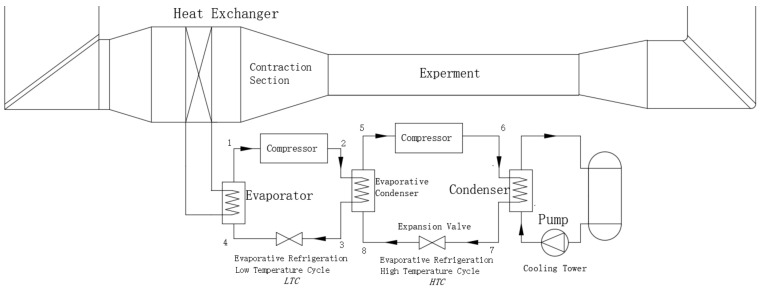
Evaporative refrigeration cycle flow.

**Figure 12 entropy-25-01312-f012:**
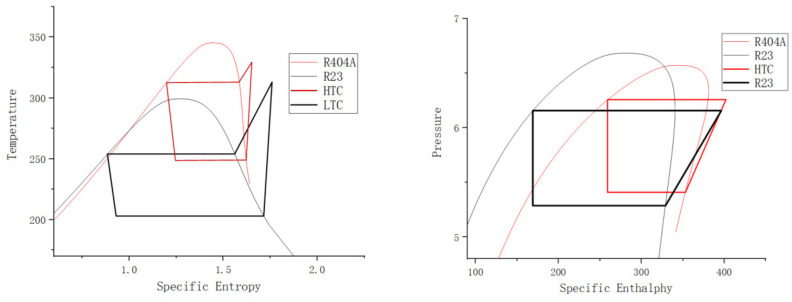
T-s and P h diagrams for the evaporative refrigeration cycle.

**Figure 13 entropy-25-01312-f013:**
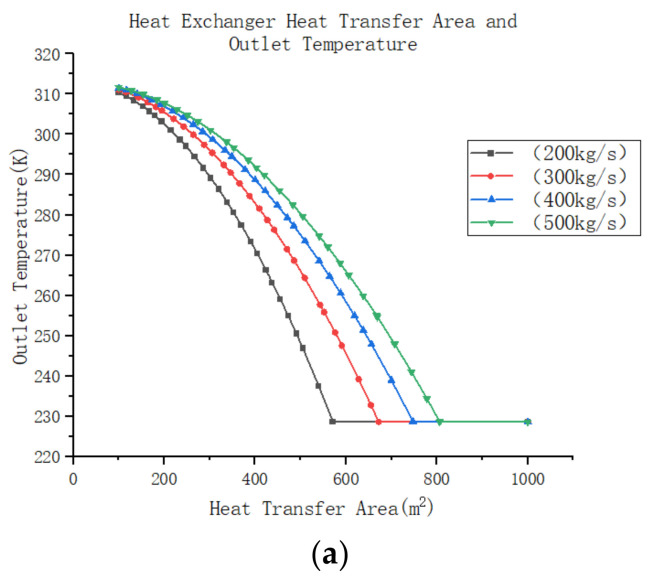
(**a**) Heat exchanger heat transfer area and outlet temperature (mass flow). (**b**) Heat exchanger heat transfer area and outlet temperature (inlet temperature). (**c**) Heat exchanger heat transfer area and outlet pressure (mass flow).

**Figure 14 entropy-25-01312-f014:**
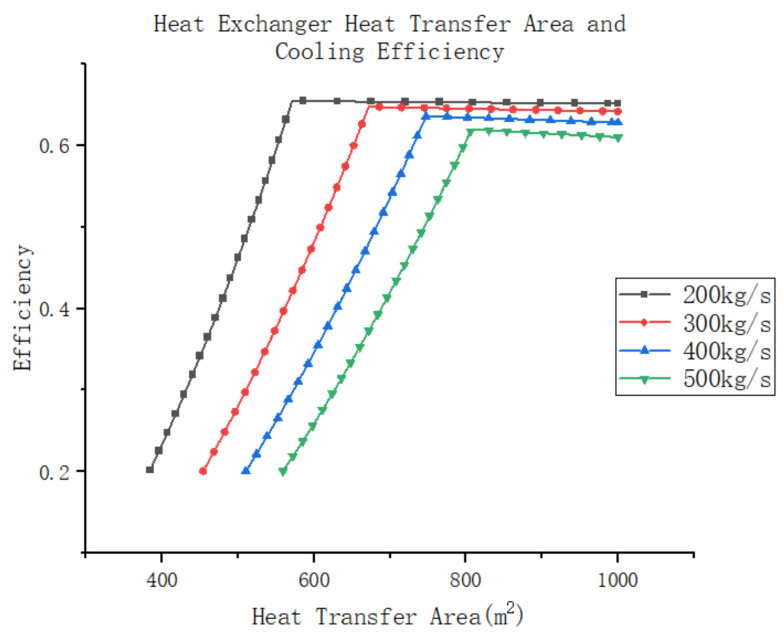
Heat transfer area and system cooling efficiency.

**Figure 15 entropy-25-01312-f015:**
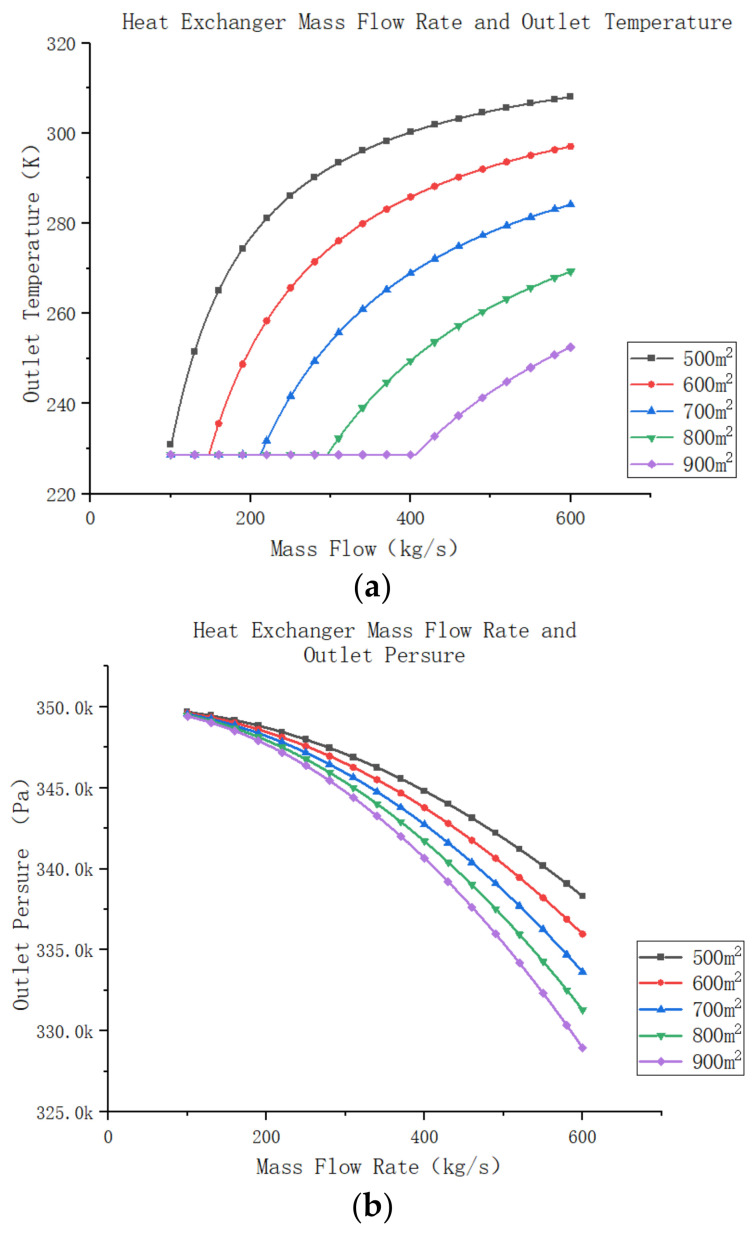
(**a**) Heat exchanger mass flow rate and outlet temperature. (**b**) Heat exchanger mass flow rate and outlet pressure.

**Figure 16 entropy-25-01312-f016:**
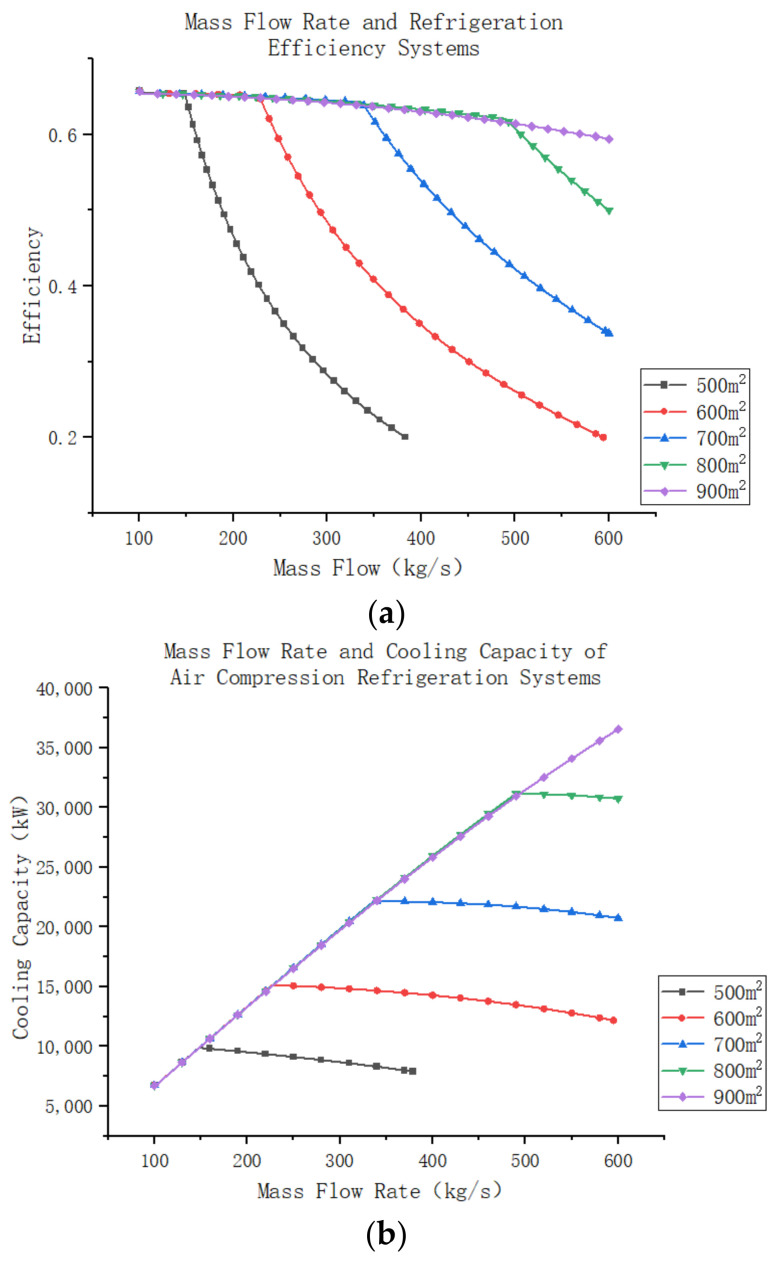
(**a**) Mass flow rate and refrigeration efficiency systems. (**b**) Mass flow rate and cooling capacity of air compression refrigeration systems.

**Figure 17 entropy-25-01312-f017:**
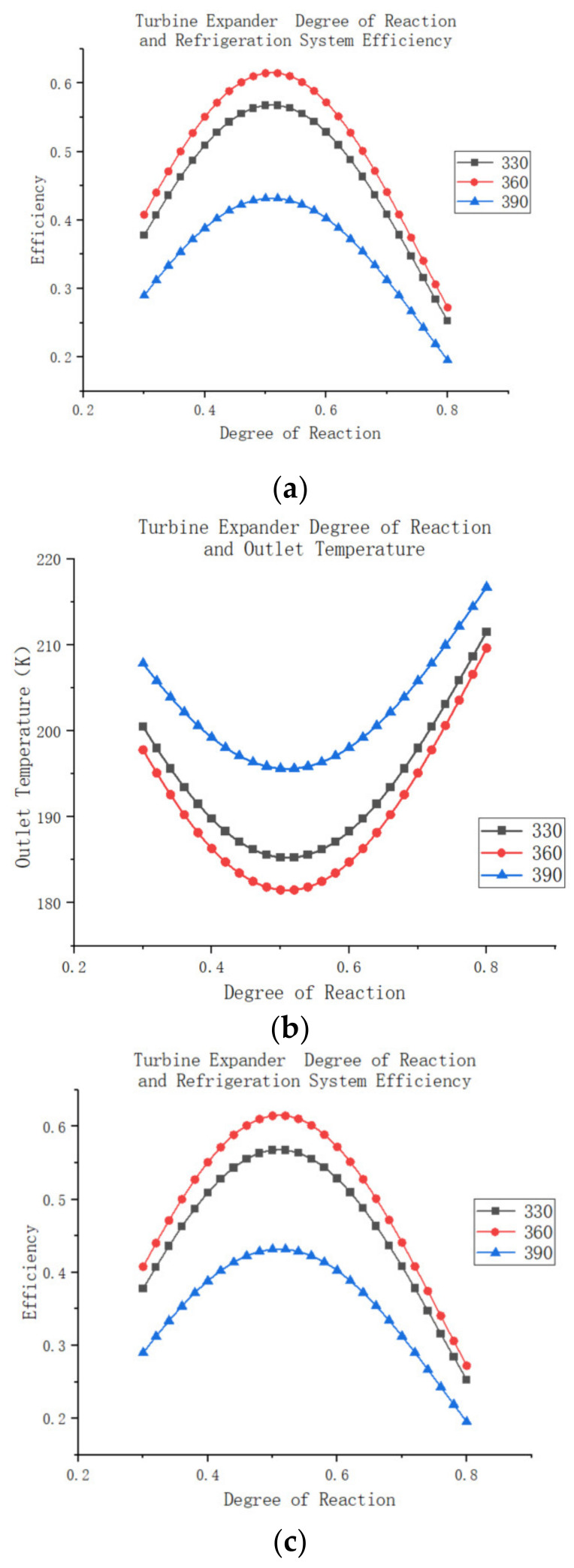
(**a**) Turbine expander degree of reaction and efficiency. (**b**) Turbine expander degree of reaction and outlet temperature. (**c**) Turbine expander degree of reaction and refrigeration system efficiency.

**Figure 18 entropy-25-01312-f018:**
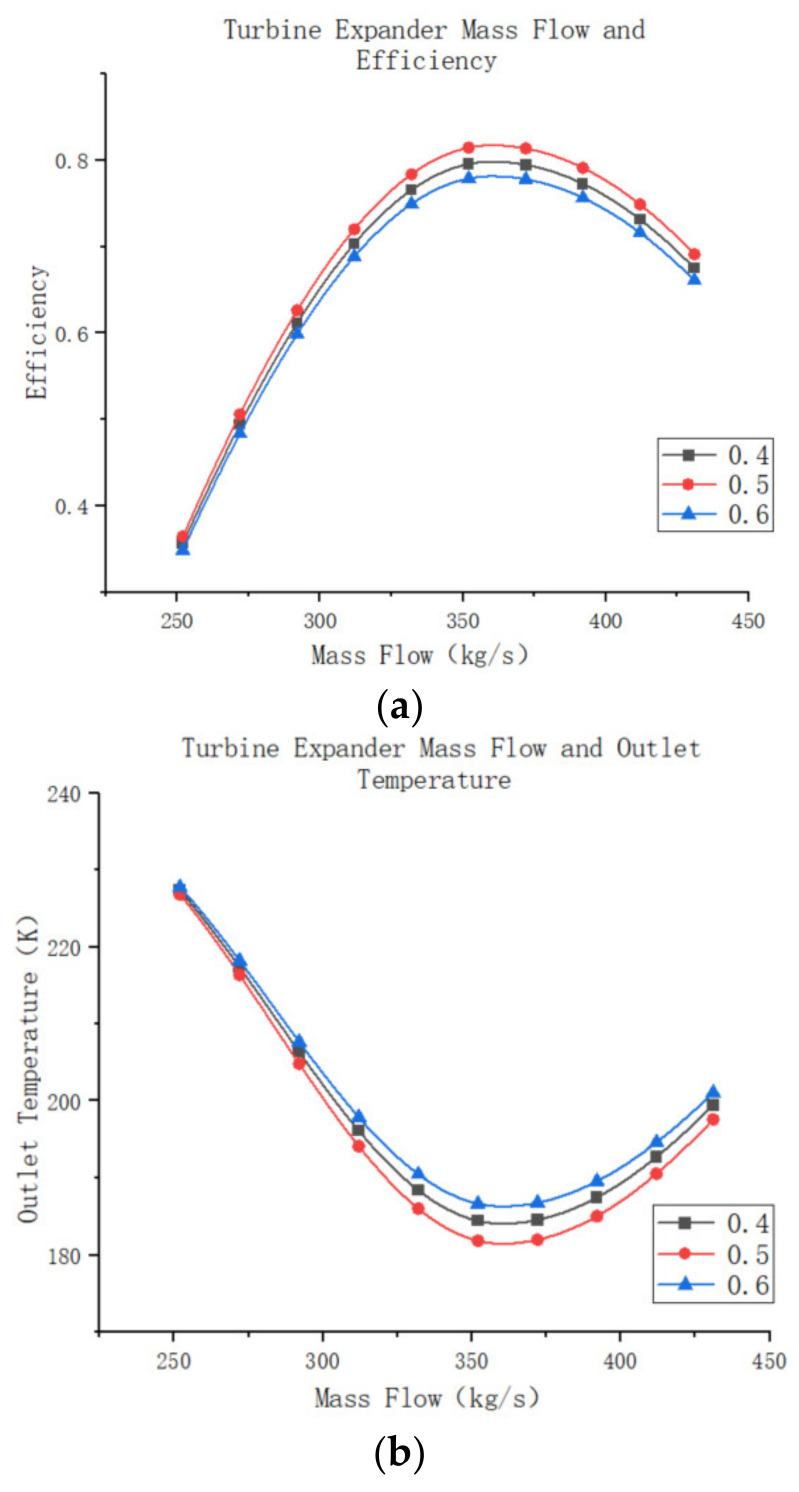
(**a**)Turbine expander mass flow and efficiency. (**b**) Turbine expander mass flow and outlet temperature. (**c**) Turbine expander mass flow and output power.

**Figure 19 entropy-25-01312-f019:**
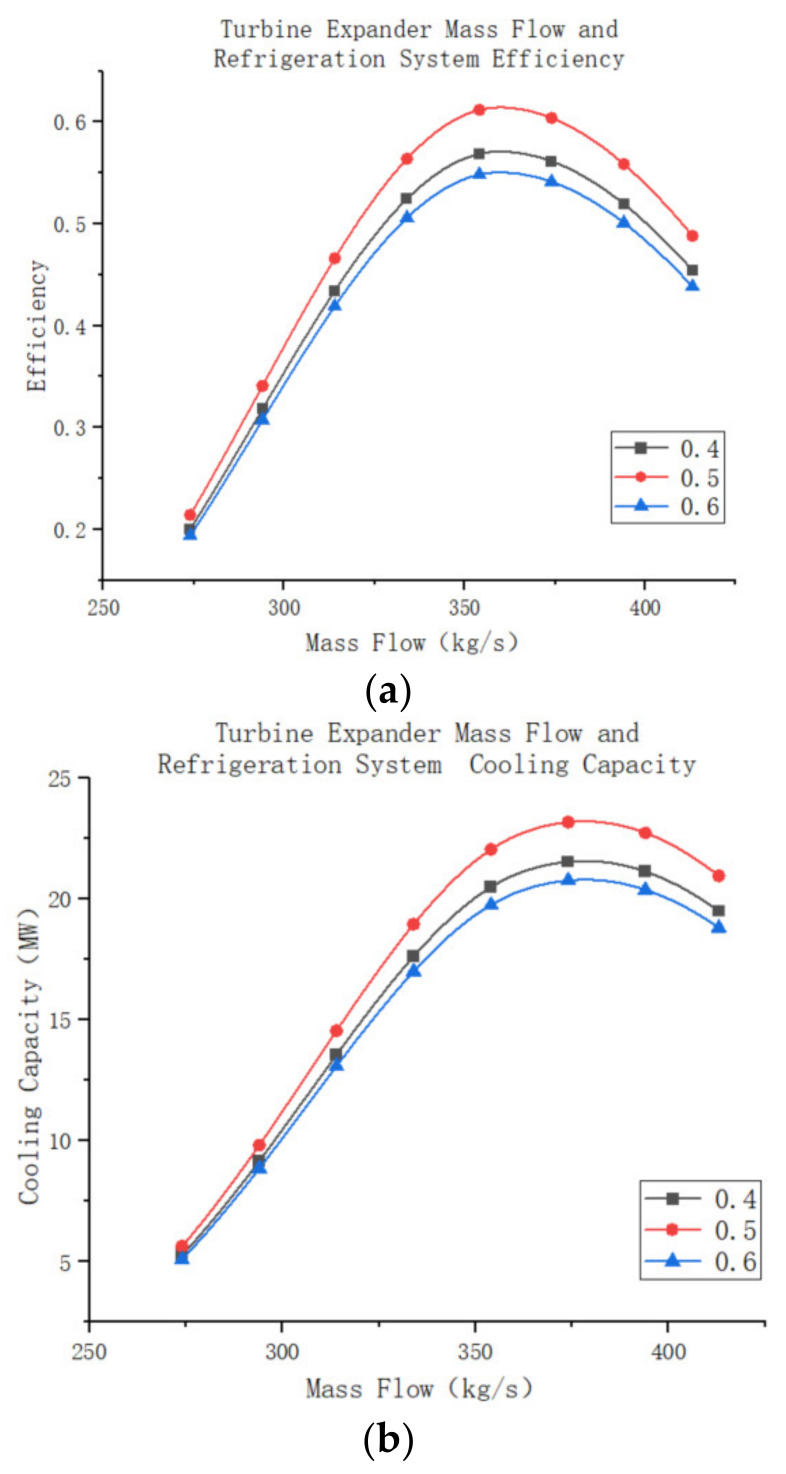
(**a**) Turbine expander mass flow and refrigeration system efficiency. (**b**) Turbine expander mass flow and refrigeration system cooling capacity.

**Figure 20 entropy-25-01312-f020:**
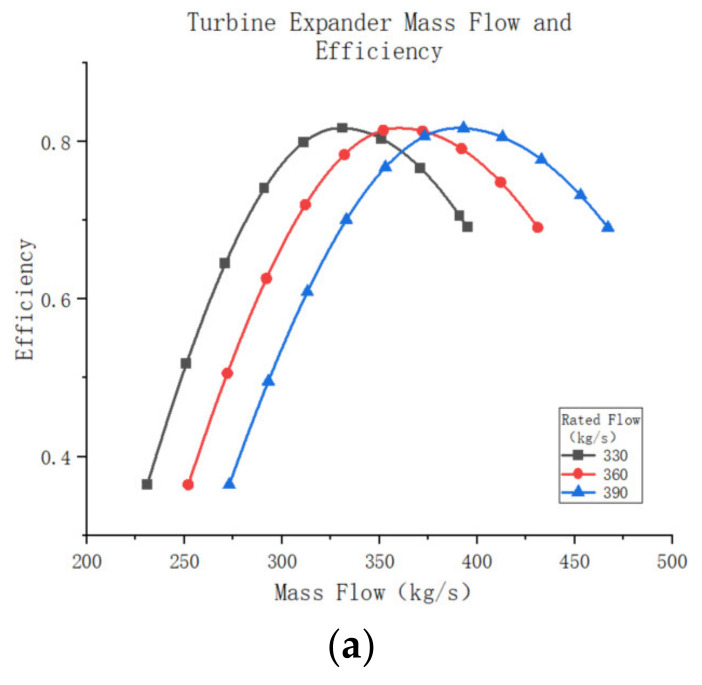
(**a**)Turbine expander mass flow and efficiency. (**b**) Turbine expander mass flow and outlet temperature. (**c**) Turbine expander mass flow and output power.

**Figure 21 entropy-25-01312-f021:**
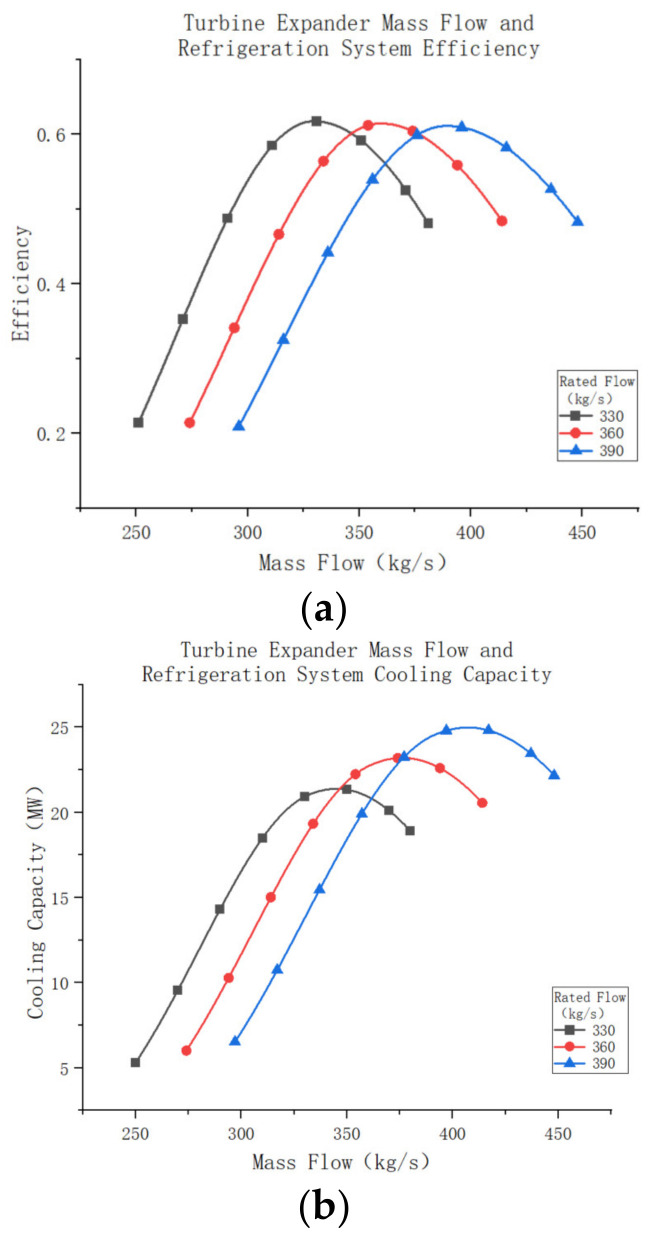
(**a**) Turbine expander mass flow and refrigeration system efficiency. (**b**) Turbine expander mass flow and refrigeration system cooling capacity.

**Figure 22 entropy-25-01312-f022:**
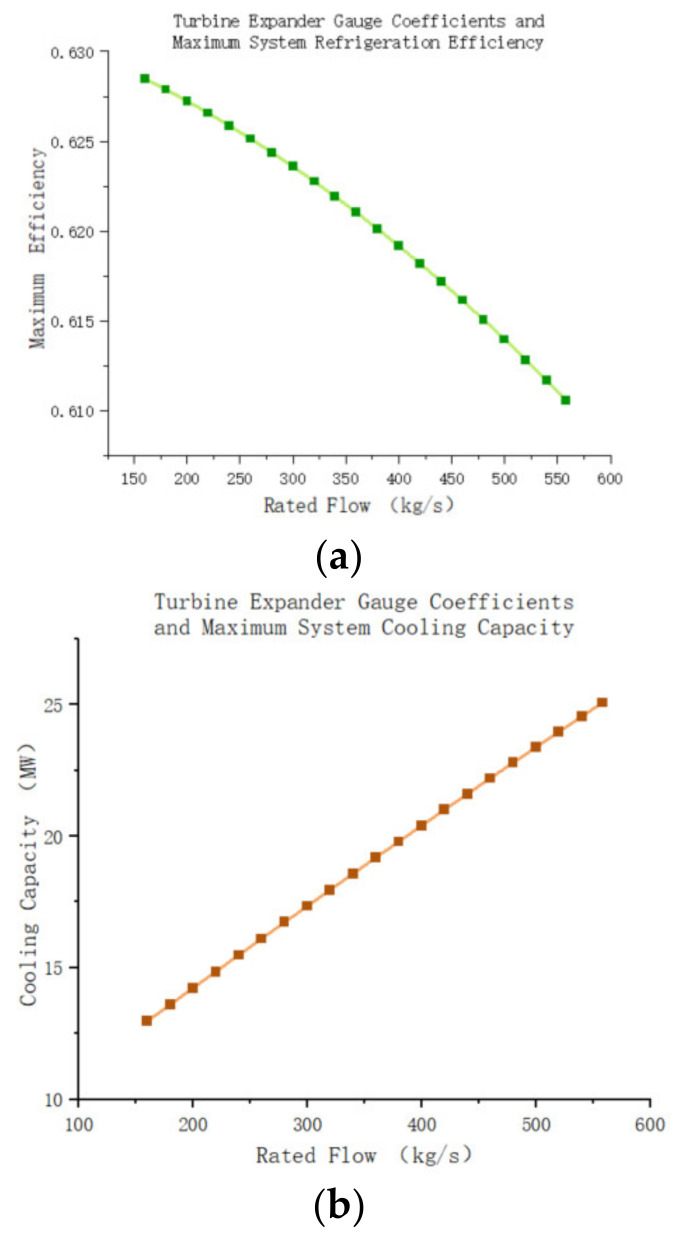
(**a**)Turbine expander gauge coefficients and maximum system refrigeration efficiency. (**b**) Turbine expander gauge coefficients and maximum system cooling capacity.

**Table 1 entropy-25-01312-t001:** Air compression cooling system component status.

Air Compression Cooling System Component Status				
	Temperature (K)	Pressure (Kpa)	Enthalpy	Specific Entropy
Centrifugal Compressors	392.55	250.00	393.56	6.87813
Compressors	456.92	414.20	459.02	6.88744
Water cooler	313.00	403.34	312.78	6.51080
Recoolers	233.00	391.05	231.83	6.22098
Expanders	152.12	110.00	151.32	6.15862

**Table 2 entropy-25-01312-t002:** Evaporative cooling system component status.

Evaporative Cooling System Component Status			
Name	Pressure (10^5^ Pa)	Temperature (°C)	Power (kW)
LTC Compressor	14.4	40.05	8427
Evaporative Condenser	14.4	−19	28,427
LTC Expansion Valve	1.94	−70	
LTC Condensers	1.94	−70	20,000
HTC Compressors	18.15	56.42	14,803
HTC Evaporators	18.15	40	43,229
HTC Expansion Valve	2.57	−24.16	
HTC Condenser	2.57	−24	28,427

Pressure and temperature conditions refer to the outlet of each component.

## Data Availability

All research data can accessed via Email junjunzh@buaa.edu.cn.

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
