# Peer review of "Designing an Environmental Wind Tunnel Cooling System for High-Speed Trains with Air Compression Cooling and a Sensitivity Analysis of Design Parameters"

_entropy, 2023, doi:10.3390/e25091312_

Round 1

Reviewer 1 Report

Designing an Environmental Wind Tunnel Cooling System for High-Speed Trains with Air Compression Cooling and a Sensitivity Analysis of Design Parameters

The manuscript presented needs several adjustments before it would publish. It is seen as confusing about the primary goal of it, and also, more explanation must be added. Right now, my suggestion is a major revision, and after the correct modifications as solicited, that paper could be published.

1. Highlights section

1.1.There are no highlights; please add them.

2. Abstract section

2.1. The novelty of the paper must be apparent. Please improve it in this section;

2.2. Numerical finds at the end of this section have to be added.

3. The introduction section.

3.1. The scientific gap that the paper has to fill must be justified;

3.2. The state of the art should be complete by adding the more recent references.

4. Thermodynamic Model for Air Compression Refrigeration Systems Section.

4.1. Please check the reference citation along the whole text; an error like this should be fixed “Error! Reference source not found.”

4.2. What do you mean by “scholarly paper”?

4.3. The “other requirements” block should be described it;

4.4. The symbols equations made improved. There are several symbols unrecognizable;

5. Modeling of Key System Components Section

5.1. A brief introduction at the beginning of the section should be made;

5.2. A flowchart of the expansion has to be added too;

5.3. I did not identify the software or computational platform the authors used to make the numerical modeling. This is mandatory, so a new subsection has to be made to explain the numerical modeling and tits assumptions;

5.4. Since it is an experimental study or a development of the practical benchmark, the authors must be added a new section explaining the components used, sensors, instrumentations, and also the uncertainties of the parameters;

5.5. Details of the benchmark must be added;

6. Results and Discussion section

6.1. A brief introduction at the beginning of the section should be made;

6.2. I believe that the results are numerical, so what about the benchmark? It is part of the paper, is not it?? I did not understand that. Please, this statement must be clarified in the abstract, title, and also introduction sections;

6.3. The results are promising, but I did not validate the results with experimental values from the benchmark or the literature, or maybe I did not understand the paper's goal. That’s why the authors must specify the goal and the activities they will present in the article.

 7. Conclusion section

7.1. Improve this section taking into account the indications above it.

Author Response

Dear reviewer:

Thank you for reviewing our article. We appreciate the thoroughness with which you assessed it and for providing valuable feedback. Your suggested improvements will undoubtedly enhance our theoretical and simulation research article. Before we respond your comments, we would like to clarify that our article does not present experimental research but rather delves into the theoretical and simulation aspects of our topic. The focus of our investigation is the scheme of the environmental wind tunnel for high-speed trains, presently in development in China. Given its enormous proportions, conducting early verification experiments is challenging. Our initial research will depend on theoretical and simulated verification instead. Concerning your recommendation, we respond as follows:

Remarks 1: Highlights section

1.1.There are no highlights; please add them.

Response 1

This suggestion is appreciated as the article may contain excessive content. This suggestion is appreciated as the article may contain excessive content. To emphasize the most important points, the inclusion of a "Highlights" section would be beneficial. Below is the added "Highlights" section - any corrections would be welcomed.

Highlights: (1)An air compression refrigeration system has been redesigned for an environmental wind tunnel for high-speed railway trains.(2) The modeling of the expander has been enhanced by using gauge coefficients

Remarks 2 :  Abstract section

2.1. The novelty of the paper must be apparent. Please improve it in this section;

2.2. Numerical finds at the end of this section have to be added.

Response 2

You are absolutely right, and we have revised the abstraction section by adding relevant content and modifying certain expressions.

Abstracts::Environmental wind tunnels for high-speed trains play a significant role in their development. The cooling system of the wind tunnel poses a challenge as it requires lower temperatures and higher cooling capacity during operation. The conventional approach to wind tunnel refrigeration uses evaporative cooling, which is less efficient at low temperatures and comes with environmental and safety risks. In this study, we propose an innovative air compression refrigeration method based on the Brayton cycle. This method converts high-pressure air into low-temperature air at atmospheric pressure for wind tunnel refrigeration. The new cooling system has reduced energy usage by 3.72MW, leading to a 13.15% improvement.The return cooler of the system is modeled using the effective number of heat transfer units and the mean temperature difference design method. Additionally, the turbine within the system is analyzed using one-dimensional flow characteristic analysis and the principle of similarity. This method has been validated by comparing it with other published papers. Subsequently, we perform a thorough sensitivity analysis on the key design parameters of the system. We observe that with a sufficient heat transfer area of the recooler, the cooling efficiency of the system exhibits a gradual decline from 64% to 60% as the mass flow rate of the system rises. For a fixed turbine, the cooling efficiency of the system rises from 20% to 62%, and subsequently declines to 37% with an increase in the mass flow rate. As a result, we conclude that the design parameters of the turbine have a more significant influence on the cooling efficiency of the system than the recooler. Our study will establish a foundation for selecting parameters to optimize the refrigeration system in the future.

Remark 3 The introduction section.

3.1. The scientific gap that the paper has to fill must be justified;

3.2. The state of the art should be complete by adding the more recent references

Response 3

We have revised the introduction accordingly and included three more recent references.  Thank you for your recommendations. We made every effort to find additional relevant articles, but there is a possibility that we overlooked some. We would appreciate any further suggestions you may have for consideration in our references.

  1. Li Y, Hu H, Sun H, Wu C. Dynamic simulation model for three-wheel air-cycle refrigeration systems in civil aircrafts[J]. International Journal of Refrigeration, 2023, 145: 353-365
  2. Qiu L,Zhang Z,Wang P. Theoretical efficiency limit of self-utilisation of compression waste heat in air separation[J]. Journal of Engineering Thermophysics, 2023, 44(7)
  3. El-Husseiny A F, Rania2;Farag, Hassan A3;Taweel, Yehia. Exergy Analysis of a Turbo Expander: Modeling and Simulation [J]. Acta Chimica Slovenica, 2022, Vol.68: 304-312.

The following excerpt comprises a portion of the revised introduction content.

The air refrigeration system used in this study was designed by Normalair Garrett Limited, a British company. They have developed a high-speed railway train air conditioning system that employs a semi-open, two-stage compressed, and boosted air refrigeration system [2]. A.J. White applied the inverse Brayton air cooling cycle to a room air conditioning system and compared its performance with and without heat rejection. The results show that increasing the heat exchanger efficiency significantly improved the system’s overall performance [3,4]. Air compression refrigeration systems are also used in the civil aircraft process, and a near-logarithm heat transfer temperature difference function is presented.[5]Furthermore, the performance of the inverse Brayton heat pump system was simulated under different operating conditions, and the differences in performance between storage and instantaneous heat modes of various heat pumps were analyzed. Zhang Han et al. [6-8] achieved electric–thermal and cooling conversion through the reverse Brayton cycle, thus enabling energy storage.Zhang analysed a 100,000 m3/h air compression system and explored methods to enhance its power.[9] Recent research on air compression refrigeration systems has resulted in improved efficiency, indicating that air compression refrigeration systems might be a viable replacement for evaporative refrigeration systems in environmental wind tunnels.

…….

Ke Changlei et al.[17] conducted numerous numerical simulations on a high-speed mixed-flow centripetal turbine under various design and cooling conditions (e.g., inlet pressure and brake power). By achieving flow capacity matching under these conditions, the method demonstrated superior performance prediction capabilities for low-temperature turbines. The air compressor is another pivotal component in air compression refrigeration systems, with energy-flow coupling characteristics akin to those of the expander. However, previous studies have focused on the performance of a single design size turbine, and rarely address how turbines of various sizes perform. El et al[18]. performed Exergy Analysis of a Turbo Expander using Matlab to investigate the performance of turbines of different sizes. This approach, however, necessitates a significant amount of experimental data and is not conducive to system design.

…….

RIOS-IRIBE [27] further explored the effects of the number of plates and plate spacing on heat transfer and pressure drop using a CFD model. Experimental validation was performed based on the simulation results, confirming the reliability of the model. This approach boasts greater precision, but it presents challenges in integrating with other models when simulating the refrigeration system. As a remedy, we will establish a numerical model to enhance computational efficiency while also incorporating various design parameters into the plate heat exchanger's performance within the system. In this paper, we aim to establish a model that incorporates the design parameters of a plate heat exchanger for analysing the performance of the exchanger as a recooler in the system.

This paper aims to introduce a novel refrigeration system, namely the air compression refrigeration system, which differs from the conventional evaporative refrigeration system used in wind tunnels. The system pressurizes dry air into high-temperature, high-pressure air and utilizes the Brayton cycle to convert it into atmospheric low-pressure gas suitable for cooling the system. Subsequently, the low-temperature gas is directly discharged into the wind tunnel, where it mixes with the original gas to achieve cooling. In previous simulations of air compression refrigeration systems, studies on turbines failed to include the analysis of turbines with different sizes, resulting in difficulties in optimizing the system selection and elevating the probability of suboptimal optimization.  To tackle this dilemma, we aim to utilize one-dimensional flow modeling and similarity principles in our study. We will establish a dynamic model of the constituents dependent on earlier research. Using this, a model of the air compression refrigeration system will be created. Through sensitivity analysis of the components, we identify the factors with the greatest influence on air compression refrigeration system efficiency, thereby paving the way for further research.

Remark 4 Thermodynamic Model for Air Compression Refrigeration Systems Section.

4.1. Please check the reference citation along the whole text; an error like this should be fixed “Error! Reference source not found.”

4.2. What do you mean by “scholarly paper”?

4.3. The “other requirements” block should be described it;

4.4. The symbols equations made improved. There are several symbols unrecognizable;

Response 4

For remark 4.1, We apologise for any inconvenience caused by our lack of attention to detail. Although the error was not noticeable on our computer, it became apparent on another device. We acknowledge that our work checking process was not thorough enough. We have now rectified the issue and are confident that it will not reoccur

For remark 4.2 “Scholarly paper” is what this article refers to. We apologize for the imprecise language. We convert this into “the article”.

For remark 4.3 We have added the introduction of "The other requirements". The other requirements include the electrical requirements of the wind tunnel, the lighting requirements, the weather simulation requirements, and life-support system.

For remark 4.4 You're right, and we have included a symbol table towards the end of the article to facilitate your comprehension of the formula

Nomenclature

p

pressure

Ω

degree of reaction

T

temperature

absolute flow angle

k

gas issentropic constant

relative flow angle

power

tip clearance

qe

specific refrigerating effect

nozzle velocity coefficient

mc

polytropic efficiency,

rotor blade velocity coefficient

q

refrigerating capacity

mass flow

Subscripts

Cp

specific heat

out

outlet

Turbine wheel diameter

in

inlet

A

heat transfer area

c

compressor

h

specific enthalphy

e

turbin

s

specific entropy 

h

recooler

l

height, m

d

cabin temperature

R

thermodynamic constant 

h

hot fluid

n

rotate speed

c

cold fluid

w

wall surfaces

Greek symbols

efficiency

expansion ratio,heat capacities

Remark 5 Modeling of Key System Components Section

5.1. A brief introduction at the beginning of the section should be made;

5.2. A flowchart of the expansion has to be added too;

5.3. I did not identify the software or computational platform the authors used to make the numerical modeling. This is mandatory, so a new subsection has to be made to explain the numerical modeling and tits assumptions;

5.4. Since it is an experimental study or a development of the practical benchmark, the authors must be added a new section explaining the components used, sensors, instrumentations, and also the uncertainties of the parameters;

5.5. Details of the benchmark must be added

Response 5

For remark 5.1 We have added a brief introduction at the begging of the chapter

The previous studies and calculations are based on conventional models of refrigeration systems. Whilst these models can determine the refrigeration performance of the system, they are not appropriate for a thorough examination of the system as the use of classical models is restricted.

    To examine the system comprehensively, it is necessary to model each individual component in detail. In an air compression refrigeration cycle, the primary components comprise the water cooler, recooler, compressor, and expander. A turbine is commonly employed as an expander in refrigeration systems. As the working process of the expander is analogous to that of the compressor, we concentrate on modeling the former to preclude redundancy. Water cooler and recooler are both types of heat exchangers. However, the primary purpose of the water cooler is to decrease the temperature of high pressure and high temperature gas, thereby maintaining system stability, and the heat exchange of the water cooler should be sufficient. In contrast, the design of the recooler is instrumental in enhancing refrigeration system efficiency. Thus, we modeled the recooler heat exchanger for analysis.

For remark 5.2 Sorry, we do not understand the expansion process being referred to. Considering our assumed turbine in the centripetal turbine, the flow of the working mass is viewed as one-dimensional. Hence, the expansion process refers to the change in air velocity.

Additionally, we have included the turbine calculation process to clarify our calculations.

For remark 5.3  This suggestion is beneficial to us. We have included the program platform and underlying assumptions.

For the calculations, we utilised Simulink in MATLAB 2018b. Our assumptions involve treating the flow of workmass in the centripetal turbine as a one-dimensional flow, disregarding the expansion process of the workmass in the worm shell and diffuser, and considering that the gas velocity does not surpass the speed of sound during flow.

Revision:

The peripheral loss of the expander is a parameter highly influenced by the operating conditions, describing the various losses generated by the working mass during the peripheral work process. These include static lobe loss, dynamic lobe loss, and residual velocity loss. In a centripetal turbine, the mass flow is typically viscous, non-constant, and three-dimensionally complex. However, in an air compression refrigeration system, where air is the sole refrigeration medium, it can be simplified as a one-dimensional, axisymmetric, adiabatic, non-viscous, and stable flow. As a result, the efficiency of the centripetal turbine can be analyzed using a one-dimensional flow analysis method[14]. At the same time, to simplify the analysis process, we consider the flow of the work mass in the centripetal turbine as one-dimensional while ignoring the expansion process of the work mass in the worm shell and diffuser. Additionally, we assume that the gas velocity is below the speed of sound during the flow process.

A one-dimensional analysis model and a turbine similarity principle code have been developed in-house by the authors using Simulink in MATLAB 2018b. The calculation procedure is illustrated in Figure 6. The initial parameters[17] are also indicated,and the geometry size of the radial-inflow turbine and the velocity triangle at each characteristic section can be calculated.

For remark 5.4 & remark 5.5  Sorry, as we previously mentioned e, this system is too extensive for us to experiment with. However, we have verified our calculations against those of others, and they perform well. This is the section on validating the method that we have recently included in the article:

Xia[13] conducted a one-dimensional flow analysis and created a three-dimensional model. The article presents a plot of the mass flow rate versus the turbine efficiency in Figure 7 when air is utilized as the fluid. The main turbine parameter specified in the article is a reaction degree of 0.51 and a blade length of 54.3mm. In Li's article [25] , a turbine with a reaction degree of 0.51 and a blade length of 8 was simulated, and relevant experiments were carried out. We can observe on Figure 8 that the results of our simulation method and the original simulation method are essentially identical through comparison.

Figure 7 Turbine expander mass flow and efficiency

 Figure 8 Comparison of the results of our code and the reference.

There might be some questions regarding the disparity between our approach and the initial approach. Compared to the original technique, our approach does not require knowledge of the absolute flow angle, relative flow angle, tip clearance, nozzle velocity coefficient and rotor blade velocity coefficient for each turbine. We only need knowledge of the degree of reaction and gauge coefficients for each turbine's. Furthermore, our approach is not only less complex but can also surmount the issue of the irrational turbine spatial structure. Additionally, we can mitigate the potential problem of gas in the turbine surpassing the speed of sound.

Remark 6. Results and Discussion section

6.1. A brief introduction at the beginning of the section should be made;

6.2. I believe that the results are numerical, so what about the benchmark? It is part of the paper, is not it?? I did not understand that. Please, this statement must be clarified in the abstract, title, and also introduction sections;

6.3. The results are promising, but I did not validate the results with experimental values from the benchmark or the literature, or maybe I did not understand the paper's goal. That’s why the authors must specify the goal and the activities they will present in the article.

Response 6

For remark 6.1 We have added a brief introduction at the begging of the chapter.

This section comprises two main parts. Firstly, we conduct a thermodynamic analysis of the environmental wind tunnel refrigeration system designed for high-speed trains, which utilises an air compression refrigeration system to ascertain its thermodynamic performance, and we compare our design with a conventional evaporative refrigeration system. Secondly, a sensitivity analysis was conducted for the key parameters in the refrigeration system using the established model. The main methodology involved determining the impact of different parameters on system performance via simulation numerical modelling.

For remark 6.2 & remark 6.3

We regret any confusion caused by the unclear structure of our paper. To showcase the effectiveness of our newly designed air compression refrigeration system in an environmental wind tunnel, we have included a comparison with results from existing literature in the previous remark 5.

Our article has two main objectives: firstly, to present our air compression refrigeration system for an environmental wind tunnel, and secondly, to conduct a sensitivity analysis on the system's parameters using numerical methods to identify the resulting optimization parameters. To better investigate the latter objective, we have made relevant improvement to our existing modeling methodology, which serves as the core concept of our paper.

Remark 7 . Conclusion section

7.1. Improve this section taking into account the indications above it.

Response 7

We have reviewed and edited the concluding section again to ensure its accuracy. And we added the following

Through the evaluation of two refrigeration cycles, it was found that the evaporative refrigeration cycle has a refrigeration efficiency of about 0.624 and requires approximately 32.1MW of external energy for operation. On the other hand, the air compression refrigeration system has a COP refrigeration efficiency of about 0.636 and needs 28.3MW of energy to function. The comparison confirms the air compression system's superiority in energy efficiency, requiring 3.72MW less energy from external sources compared to the evaporative refrigeration system. The air compression system consumes 3.72 MW less external energy to operate compared to the evaporative refrigeration system, with a 13.15% improvement in efficiency.

We hope our response meets your expectations. Yours advice are very helpful, thank you for your advice again. Please do not hesitate to let us know if you have any further suggestions in the future.

Kind Regards,

Yours sincerely,

Junjun  Zhuang

Reviewer 2 Report

Technical issues with figure cross-referencing persist, displaying the error

”Error! Reference source not found.” across multiple instances. This hampers

the reader’s understanding and connection with the referred figures.

Manuscript ID: entropy-2576471

Title: “Designing an Environmental Wind Tunnel Cooling System for High-

Speed Trains with Air Compression Cooling and a Sensitivity Analysis of Design

Parameters”

Authors: Zhuang Junjun, Liu Meng, Wu Hao, and Wang Jun

This manuscript explores the integral role of environmental wind tunnels in

high-speed train development. The authors propose an air compression refrigeration

technique rooted in the Brayton cycle as a solution to challenges faced

by the conventional evaporative cooling method.

However, there are significant areas of concern:

• The paper lacks experimental validation. The findings, while theoretically

insightful, are presented without any corroborating experimental evidence,

raising questions about the reliability and real-world applicability of the

data.

Author Response

Dear reviewer:

Thank you for acknowledging our dissertation. Your feedback on our paper was invaluable in improving its quality. We appreciate your assistance in this matter.

We have carefully considered your comments and made the necessary changes to the essay accordingly.

Remarks 1: The paper lacks experimental validation. The findings, while theoretically insightful, are presented without any corroborating experimental evidence, raising questions about the reliability and real-world applicability of the data.

Response 1

Your comment is insightful, but its experimental validation may pose a challenge. Our simulation involves a large system, and its aim is to offer guidance to subsequent experiments and minimize the investment in them.

Furthermore, we employed the classical algorithm in the MATLAB environment to address the effectiveness of the compressed air refrigeration system,and we used the software Solkane to calculate the evaporative cooling algorithm.For the calculation method of our subsequent sections, we utilise the traditional algorithm for the heat exchanger part and apply the Matlab Simulink for the calculation process.

We have incorporated a comparison between our approach and experimental findings from other published papers in the article to validate our One-dimensional flow characteristics analysis and similarity principle. This additional segment emphasises the practicality of our method.

A one-dimensional analysis model and a turbine similarity principle code have been developed in-house by the authors using Simulink in MATLAB 2018b. The calculation procedure is illustrated in Figure 6. The initial parameters[17] are also indicated,and the geometry size of the radial-inflow turbine and the velocity triangle at each characteristic section can be calculated

Figure 6 Turbine modeling flowchart

Xia[13] conducted a one-dimensional flow analysis and created a three-dimensional model. The article presents a plot of the mass flow rate versus the turbine efficiency in Figure 7 when air is utilized as the fluid. The main turbine parameter specified in the article is a reaction degree of 0.51 and a blade length of 54.3mm. In Li's article [25] , a turbine with a reaction degree of 0.51 and a blade length of 8 was simulated, and relevant experiments were carried out. We can observe on Figure 8 that the results of our simulation method and the original simulation method are essentially identical through comparison.

Figure 7 Turbine expander mass flow and efficiency

 Figure 8 Comparison of the results of our code and the reference.

Remarks 2 : Technical issues with figure cross-referencing persist, displaying the error ”Error! Reference source not found.” across multiple instances. This hampers the reader’s understanding and connection with the referred figures.

Response 2

We apologise for any inconvenience caused by our lack of attention to detail. Although the error was not noticeable on our computer, it became apparent on another device. We acknowledge that our work checking process was not thorough enough. We have now rectified the issue and are confident that it will not reoccur.

Remarks 3:The presentation of 21 standalone fgures in the results section, each containing a singular plot, is excessive and not typical. Consolidating related plots into fewer fgures would enhance the paper’s presentation and readability.

Response 3

This suggestion is helpful to us. We have rearranged the numbering of the pictures and grouped together those that depict the same simulation state. This enables us to express our ideas more clearly. We have provided our original paper and would welcome your further feedback.

We hope our response meets your expectations. Yours advice are very helpful, thank you for your advice again. Our article has been extensively revised,and please do not hesitate to let us know if you have any further suggestions in the future.

Kind Regards,

Yours sincerely,

Junjun  Zhuang

Round 2

Reviewer 1 Report

The paper is ready to be published

Reviewer 2 Report

Figures 13-a,b, and c can be combined under one caption, and labels (a), (b), and (c) can be mentioned below the figures. Similarly, for other figures. Please have a look at some recently published papers to get som idea about this. The manuscript can be accepted.

NA